# Life Cycle of Stratocumulus Clouds over one Year at the Coast of the Atacama Desert

Jan Herbert Schween[1], Camilo del Rio[2,3], Juan-Luis García[2,3], Pablo Osses [2,3], Sarah Westbrook[1], and Ulrich Löhnert[1]

[1] Inst. f. Geophysics and Meteorology, Universtity of Cologne, Germany
[2] Inst. de Geografia Pontificia Universidad Católica de Chile, Santiago, Chile
[3] Centro UC Desierto de Atacama, Pontificia Universidad Católica de Chile, Santiago, Chile

**Correspondence:** Jan H. Schween (jschween@uni-koeln.de)

**Abstract.** Marine stratocumulus clouds of the Eastern Pacific play an essential role in the Earth's energy and radiation budget. Parts of these clouds off the west coast of South America form the major source of water to the hyper-arid Atacama Desert coastal region at the northern coast of Chile. For the first time, a full year of vertical structure observations of the coastal stratocumulus and their environment are presented and analysed. Installed at Iquique Airport in northern Chile in 2018/2019, three state-of-the-art remote sensing instruments provide vertical profiles of cloud macro- and micro-physical properties, wind, turbulence and temperature, as well as integrated values of water vapor and liquid water. Distinct diurnal and seasonal patterns of the stratocumulus life-cycle are observed. Embedded in a land-sea circulation with a super-imposed southerly wind component, maximum cloud occurrence and vertical extent occurs at night, whereas minima during local noon. Night-time clouds are maintained by cloud-top cooling, whereas afternoon clouds re-appear within a convective boundary layer driven through local moisture advection from the Pacific. During the night, these clouds finally re-connect to the maritime clouds in the upper branch of the land-sea circulation. The diurnal cycle is much more pronounced in austral winter with lower, thicker and more abundant (5x) clouds than in summer. This can be associated to different SST gradients in summer and winter, leading to a stable, respectively neutral stratification of the maritime boundary layer at the coast of the Atacama Desert in Iquique.

## 1 Introduction

Stratocumulus clouds cover, averaged over the year, about 23% of the oceans and 12% of the land surface, making them the most abundant cloud type (e.g. Wood, 2012). As they reflect between 30% and 60% of the incoming solar radiation they have a cooling effect and play an important role in the radiation budget of the planet. Large stratocumulus cloud fields can be typically found at the western coasts of the continents where equatorward cold ocean currents like the Humboldt- and Benguela Current meet stable atmospheric stratification mediated by the subsiding branch of the Hadley Circulation. Under these conditions, a persistent stratocumulus cloud deck forms with a mixed maritime boundary layer (MBL) and an extremely sharp inversion above, separating the cloud layer from the free troposphere. This system is stabilized by several feedback mechanisms: Radiative and evaporative cooling at cloud-top initiate turbulent mixing between cloud and ocean surface. Evaporation from the ocean and mixing through the boundary layer provides a continuous flow of water vapor balancing the water loss at cloud-top

(e.g. Schubert et al., 1979; Stevens et al., 2003) and precipitation back into the ocean (Wood, 2012). It has been hypothesized that evaporative cooling at cloud top may increase turbulent exchange between the moist boundary layer and the dry, free troposphere and thus dissipate the cloud (cloud top entrainment instability, CTEI). But it has been found that this mechanism requires special conditions and occurs less frequent than originally thought (Wood, 2012). The stratocumulus in the southeast Pacific off the coast of Peru and northern Chile has been identified to have the largest geographical extent and to be the most persistent of the world (Klein and Hartmann, 1993). In addition to their global role, these clouds provide fresh water to coastal ecosystems in the form of fog in deserts like the Atacama Desert (e.g. Muñoz-Schick et al., 2001; Cereceda et al., 2008a; Manrique, 2011). Due to the extreme aridity in the Atacama, this is the only significant water source and, accordingly, it determines several processes from genetic evolution to surface mineral formation (Dunai et al., 2020).

The mechanisms behind the formation and persistence of maritime stratocumulus have been investigated in many model studies from the first bulk layer model of Lilly (1968) and its improvement by including moist thermodynamics (Schubert et al., 1979) or allowing a decoupling of the ocean from the cloud (Turton and Nicholls, 1987). These bulk models rely on basic physical principles like conservation of energy and matter. They assume a well mixed layer between surface and cloud-top and need parameterizations of the turbulent surface and entrainment fluxes. Global or regional circulation models cannot resolve structures of the stratocumulus clouds nor the microphysical processes within the cloud, but they allow to investigate their role in the local or global climate. An investigation of stratocumulus clouds in models of the Coupled Model Intercomparison Project (CMIP5) revealed that these state of the art climate models have problems to represent stratocumulus above the southeast Pacific correctly (Lin et al., 2012). When comparing with observations, these models showed a too low cloud cover, too high cloud-tops and non realistic cloud reflectivities. Reasons for these shortcomings were identified as the turbulence driven by radiative cooling at cloud-top which is not represented in most of the models, as well as the sharp inversion at the top of the maritime boundary layer (MBL) which cannot be resolved by these models (Lin et al., 2012, and references therein).

The hope would be that the latter problems could be solved by large eddy simulations (LES). These models resolve the large turbulent eddies mediating the transport in the MBL and parameterize the small scale turbulence based on the structure of these large scale eddies. A study comparing several LES models against observations showed that these models overestimated the turbulent mixing and thus entrainment at cloud-top (Stevens et al., 2005). As a result, water contents were too low and the cloud decoupled from the underlying MBL. The authors identified as reason for the too strong entrainment, a too low (vertical) resolution at cloud-top and insufficient understanding of subgrid scale physics in this very region. Despite these difficulties, LES models can reveal insight into processes which cannot be resolved with global circulation models. E.g., Schneider et al. (2019) show that a drastic increase of $CO_2$ could lead to a breakup of the large stratocumulus fields above the oceans and that this break up could only be reversed when returning to pre-industrial $CO_2$ levels.

As even LES models cannot represent all aspects of stratocumulus cloud processes, one has to rely on observations. The stratocumulus at the west coast of North America has been investigated during several campaigns from ships and airplanes like the Dynamics and Chemistry of Marine Stratocumulus (DYCOMS) field studies (e.g. Stevens et al., 2003) which focused on the dynamics, especially the entrainment at cloud-top. Measurements were made from airplanes during dedicated days. The

Marine ARM GCSS Pacific Cross-Section Intercomparison (GPCI) Investigation of Clouds (MAGIC) field campaign used remote sensing instrumentation on a ship to investigate the structure of the stratocumulus cloud deck off the west coast of North America, i.e. its longitudinal gradient (Zhou et al., 2015).

The EPIC, PACS Stratus 2003 and 2004 campaigns (Serpetzoglou et al., 2008; Bretherton et al., 2004) as well as the VAMOS Ocean-Cloud-Atmosphere-Land Study Regional Experiment (VOCALS-REx) used a whole set of observations from maritime platforms like ships, airplanes and satellites to investigate the structure of the southeast Pacific stratocumulus west of the coast of Peru and northern Chile (Wood et al., 2011; Mechoso et al., 2014). The goal of these campaigns was to increase the understanding of the coupling between ocean surface and atmosphere and cloud-aerosol-precipitation interactions. All these campaigns were limited in time.

To investigate the long term development of the stratocumulus of the southeast Pacific (Schulz et al., 2012; Muñoz et al., 2016) analysed airport observations of cloud-base from the three coastal Chilean airports of Arica (18.5°S), Iquique (20.5°S) and Antofagasta (23.7°S). Cloud occurrence shows a clear diurnal and seasonal pattern with maxima during night in winter and spring. While Schulz et al. (2012) found a weak decrease in cloud cover, Muñoz et al. (2016) could show that with higher temporal resolution and restriction to low clouds only, a long term increase in spring and decrease in fall as well as a cloud base decrease of 100 m per decade can be found. A similar pattern in cloud cover can be found in satellite data with a weak increase in cloud cover in winter and spring especially over the ocean, and a decrease above land with altitudes above 1000 m asl (del Rio et al., 2021a). Besides these weaker long term trends, cloud cover shows a strong inter-annual variability which is connected to the El Niño Southern Oscillation (ENSO) global circulation pattern with opposite effects in different seasons. During an El Niño phase with warm equatorial waters, cloud cover is, in comparison with other years, increased in summer and decreased in winter and vice versa in La Niña years (del Rio et al., 2021a).

Up to now, no continuous, long-term observations of the vertically resolved dynamic, thermodynamic and microphysical properties of these coastal stratocumulus and their environment exist. These parameters are important for quantifying heating rates, understanding the cloud life cycle and would provide valuable information for model evaluation and development of parameterization schemes. We present here data from a one year deployment of three state-of-the-art remote sensing instruments at the airport of Iquique (Chile, 20.54°S, 70.18°W) as part of the German Research Foundation (DFG) funded collaborative research center 1211 'Earth - Evolution at the Dry Limit' (https://sfb1211.uni-koeln.de) in close cooperation with Centro del Desierto de Atacama (Pontificia Universidad Católica de Chile, UC).

The main objective of this paper is to quantify and understand the very pronounced diurnal cycle of these coastal stratocumulus clouds, and relate it to the seasonal differences in sea surface temperature (SST) and the large scale flow. In addition, we investigate mechanisms which transport water vapor and heat into, or out of the cloud. Specifically, stratocumulus clouds show strong evaporation at their top and may loose liquid water by drizzle. The related water loss must be compensated by transport of water vapor from the ocean surface. The interplay of these processes is still not very well understood (see Wood, 2012) and our data set provides insights through revealing details of the atmospheric boundary layer below the stratocumulus.

The paper is structured as follows: Section 2 describes the Iquique airport site as well as the deployed instrumentation. The retrievals used to derive the atmospheric and cloud properties are shortly described. Section 3 presents statistics of the

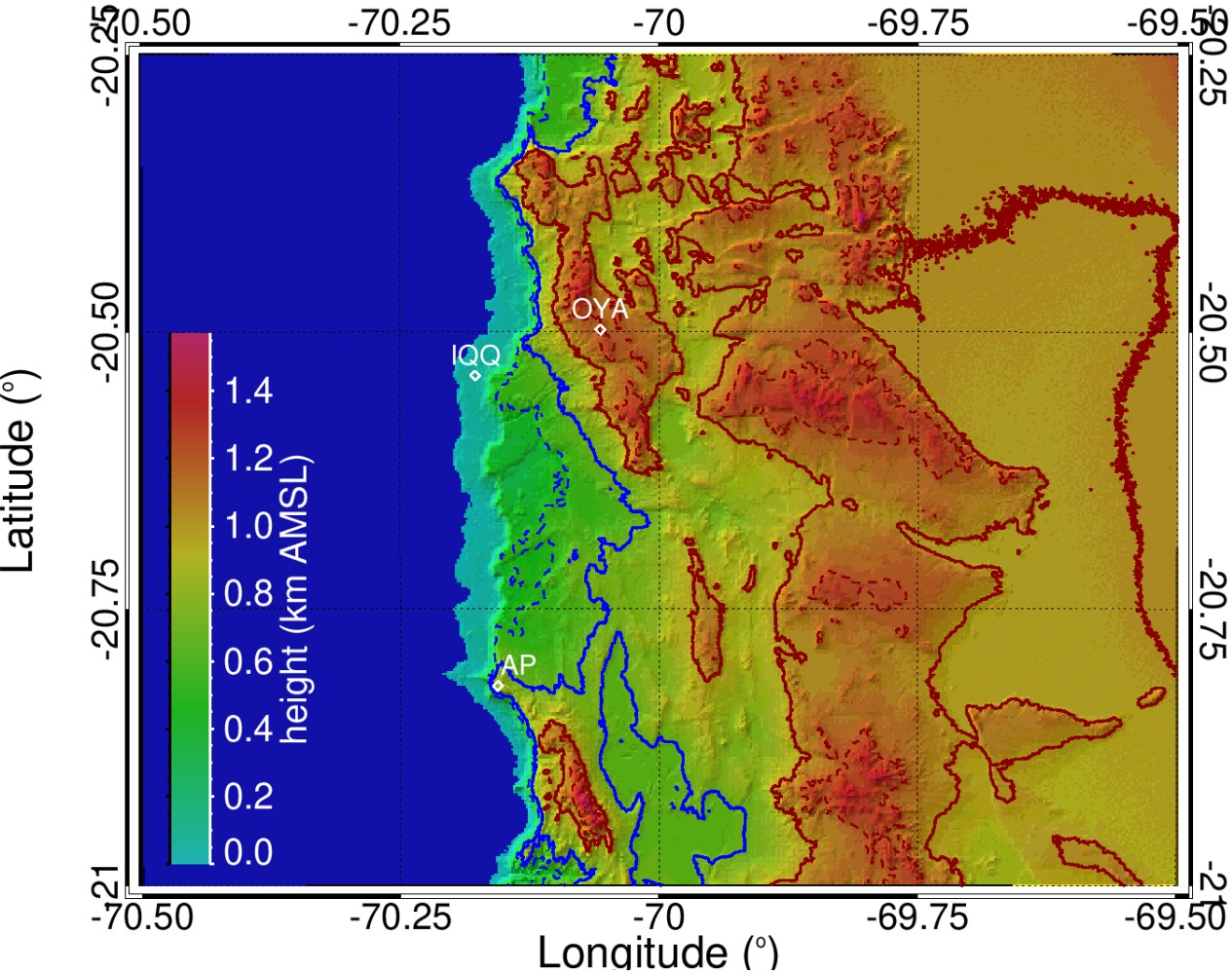

**Figure 1.** Topography map of the region around the measuring site at Iquique Airport. IQQ depicts the location of the instruments, OYA is the position of the main meteorological station of the Oyarbide site, AP is the Alto Patache UC desert research station. Lines indicate the elevation range of the stratocumuls cloud as we observed it in this study from IQQ. Blue and red lines show cloud-base and cloud-top height, respectively. Solid lines indicate the average and dotted lines the average ± standard deviation.

obtained observations in order to characterize diurnal and yearly variability of the stratocumulus and finally section 4 provides

a scientific discussion of the observed stratocumulus life cycle at the Atacama Desert coast line.

## 2 Measurement Configuration

### 2.1 Measurement Site Iquique

The instruments were deployed from March 2018 to March 2019 on grounds of the airport of Iquique at the coast of northern Chile at 20°32'22"South, 70°10'38"West about 2.5 km from the coast line (Fig. 1). The period is characterized by a return from a rather weak La Niña phase ending in April 2018 (Oceanic Niña index ONI = -0.7) and an comparably weak El Niño phase beginning in October 2018, lasting until June 2019 (average ONI=0.7 with maximum ONI=0.9 around November 2018, see NOAA Climate Prediction Center, 2022). The airport is located on flat terrain at about 56 m above sea level (asl) between the ocean and the coastal mountain range, which forms here an about 400 m high cliff that lies 4 km from the coast line. The site is especially characterized by its location on the northern end of a coastal plain which extends about 30 km to the south, is 2 km to 4 km wide and between 20 m and 50 m above the ocean. This a somewhat unique setting at the coast of northern Chile, as the cliff usually drops from several hundred meters directly into the ocean with only narrow stretches of rocky beaches. The cliff is part of the coastal mountain range with further inland heights around 1100 m and summits around 1400 m asl. The stratuscloud thus interferes with the topography whithin a few kilometers, provides as fog water to the surface and is limited in its extension to the east (blue and red lines in Fig.1). About 120 km to the east rise the Andes to heights above 5000 m forming the western border of the Altiplano, a plateau at 3800 m with the salt pan of the Salar de Uyuni. Between the Andes and the coastal mountain range lies a central valley at heights around 1000 m. In the area between the coast and the Andes lies the Atacama desert.

Besides the logistic advantages of an airport (electricity, communication and safety) an additional reason for the choice was the vicinity to the Oyarbide site (operated by Pontificia Universidad Católica UC and Heidelberg University) a few km inland with standard meteorology and fog collectors (García et al., 2021; del Rio et al., 2021b). Standard meteorological observations from the airport were used in Schulz et al. (2012) and Muñoz et al. (2016) to evaluate long term trends of stratocumulus clouds in the eastern Pacific.

The airport of Antofagasta about 320 km to the south hosts a radiosonde station (WMO ID 85442). Launch is every day at 12UTC, i.e. about 2.5 hours after sunrise. Data from these soundings has been used to derive statistical retrieval algorithms for the deployed microwave radiometer (Section 2.2.2). Three remote sensing instruments were deployed: a microwave radiometer (MWR), a cloud radar (CR) and a doppler wind lidar (DL). Additionally, two simple meteorological stations at heights of 1 m and 2 m above ground were installed.

### 2.2 Instruments

#### 2.2.1 Radar

The cloud radar is a Frequency Modulated Continuous Wave (FMCW) Radar RPG-FMCW-94 manufactured by RPG (Meckenheim Germany, Küchler et al., 2017). The instrument was installed so that the radar beam is oriented vertically. The frequency of the emitted radar waves is modulated around 94 GHz in a saw-tooth pattern resulting in reoccurring so-called chirps. In the

**Table 1.** Main parameters of the chirps used by the cloud radar. Hmin, Hmax and Hres are minimum, maximum and resolution of the height, vrange and vres are velocity range and resolution, respectively, and intL is the integration time over each chirp.

| chirp# | Hmin | Hmax | Hres | vrange | vres | intL |
|--------|------|------|------|--------|------|------|
| | m | m | m | m s$^{-1}$ | m s$^{-1}$ | s |
| 1 | 100 | 900 | 11.2 | $\pm9.0$ | 0.018 | 0.547 |
| 2 | 900 | 3700 | 22.7 | $\pm7.2$ | 0.028 | 1.084 |
| 3 | 3700 | 12000 | 18.8 | $\pm3.2$ | 0.013 | 1.760 |

radar software, the user can define the dependent parameters height range and resolution as well as velocity range and reso-
lution, which determine the chirp parameters for the instrument. To cover the height range from 0.1 km to 12 km we defined
three different chirps. Height ranges of the different chirps were adapted to the expected heights of the stratocumulus. Each
of the chirps is repeated between 6000 and 9800 times leading to a total integration time of 3.39 s before the chirp sequence
is repeated. Height and velocity ranges and resolutions as well as integration time per chirp are depicted in Tab. 1. From the
backscattered and received signal, the CR derives Doppler velocity spectra, radar reflectivity factor $Z_e$, mean and standard
deviation as well as skewness of the Doppler velocity spectrum.

### 2.2.2 Microwave Radiometer

The microwave radiometer is a HATPRO (Humidity And Temperature PROfiler, Rose et al., 2005) manufactured by RPG
GmbH (Meckenheim, Germany). It measures thermal radiation as brightness temperatures (TB) in the microwave range in
seven channels at the oxygen emission line complex at 60 GHz (W-Band) and seven more channels at the higher flank of the
22.2 GHz water vapor emission line (K-band). Zenith TB are measured every second. From these brightness temperatures inte-
grated water vapor (IWV), liquid water path (LWP), temperature profiles and water vapor profiles are derived. The instrument
was calibrated against liquid nitrogen at the beginning of the campaign. To ensure stability, the instrument performs frequent
internal calibrations against a noise diode and a hot load kept at a controlled temperature.

Retrievals to derive the atmospheric variables from the TB are based on multi-variate regressions based on the 14 TB
channels (Löhnert and Crewell, 2003; Crewell and Löhnert, 2007). A multi-year radiosonde data set is used to simulate the
14 TB channels via a line-by-line radiative transfer model. Specifically, we used 20 years of data from the nearest radiosonde
in Antofagasta (4297 profiles from 1998-2018, i.e. 67% of all data after a thorough quality control). The simulated TBs are
related to the corresponding atmospheric variables using a quadratic minimization. The resulting regression coefficients are
then used to derive the atmospheric variables from TB measurements at Iquique.

To improve the accuracy of the temperature profiles, so-called boundary layer scans are performed every 15 minutes. These
scans are performed in the vertical plane at 70° azimuth, and consist of 19 elevation angles with nine elevations from 4.2° to
30°, a zenith observation and symmetrical elevations on the opposite side. The scan ran thus from a direction to the Oyarbide
measuring field towards the ocean. The idea was to eventually investigate differences in stratification towards the ocean and

towards the mountains. Unfortunately, the lowest elevation angles were lower than the line of sight towards the cliff and the inland half of the scan could not be analysed meaningfully. Due to this reason, only the scans towards the ocean were used for this enhanced temperature profiling. These boundary layer scans allow derivation of temperature profiles with low uncertainty (<0.5 K RMSE) in the lowest hundred meters increasing further up to the middle troposphere, (1.5 K RMSE, see Crewell and Löhnert, 2007). Water vapor profiles are vertically coarsely resolved using MWR measurements, i.e. there are only approximately two independent pieces of information contained in the TBs for water vapor profiling (Löhnert et al., 2009).

An intrinsic feature of passive remote sensing is that spatial resolution decreases with distance. As a result the inversion appears much broader than it is in reality, and it becomes impossible to identify bottom and top of the inversion layer. It will be also difficult to identify a moist adiabatic temperature profile in a cloud layer of a few hundred meters directly below the inversion. To estimate the height of the inversion we first identify the maximum potential temperature gradient, fit a second-order polynomial to the three gradient values around and determine the location of the maximum of this polynomial as the height of the inversion.

LWP is not directly measured by the radiosonde used for calibration, but must be diagnosed from relative humidity and temperature. It is assumed that condensation occurs and a cloud forms when relative humidity $RH$ exceeds a threshold $RH_{\mathrm{cld}}$. Liquid water content in the cloud is assumed to follow a modified moist adiabat (Karstens et al., 1994) and is finally integrated to provide LWP. This method can introduce some systematic LWP offsets in comparison to real clouds, especially if thin cloud layers like stratocumulus are present. Additionally, the vertical resolution of radiosonde profiles from Antofagasta prior to about the year 2000 is very low. As a result, the standard threshold of $RH_{\mathrm{cld}} = 95\%$ lead to complete years diagnosed with no or only few clouds. A comparison of the vertical frequency distributions of clouds from the Cloudnet classification scheme derived from our 2018/2019 data (Section 2.3.1) with a cloud distribution from the radiosonde based on different $RH$ thresholds revealed that for every year a different threshold would be necessary. Given the uncertainty of this method, we decided to calculate LWP with four different retrievals based on the four thresholds 80% (TH80), 85% (TH85), 90% (TH90), and 95% (TH95) in $RH$. The differences between the four thresholds thus give a measure for the uncertainty for the special situation at the Iquique airport site.

### 2.2.3 Doppler Lidar

The Doppler wind lidar deployed is a Streamline XR from HALO Photonics (Worcestershire, Great Britain). It sends laser pulses at a wavelength of 1500 nm at a rate of 10 kHz into the atmosphere, measures the backscattered signal, integrates over one second and derives the along-beam mean Doppler velocity and backscatter coefficient at 30 m spatial resolution. Maximum range is 15 km, but the Doppler signal detection needs sufficiently strong backscatter which typically limits the applicable range to the atmospheric boundary layer with higher aerosol contents. The instrument performed a conical scan at an elevation of 70° every 15 minutes and at 24 azimuth angles (VAD scan) to derive a vertical profile of the horizontal wind vector based on the method described in Päschke et al. (2015). This scan was followed by a scan in the same vertical plane as the MWR (RHI scan). During the remaining time between scans, it measured vertical velocities at a temporal resolution of one second to characterize the turbulence in the boundary layer. These vertical measurements sum up to 52min 44sec per hour and

are used for an uncertainty estimate of the horizontal winds as well as for the boundary layer classification scheme described in Section 2.3.2. Vertical backscatter measurements are used to derive cloud base (Section 2.3.1).

### 2.2.4 Standard meteorology

Both the MWR and the CR are equipped with a standard weather station (WXT 530, Vaisala, Finland). The station measures air pressure, wind speed and direction, temperature and relative humidity as well as precipitation, providing values every 1 sec (MWR) and 4 sec (CR), respectively. The two stations are mounted at 2.25 m and 1.25 m and thus give information about the near-surface stratification.

Intercomparison of air temperatures ($T_a$) from our instruments and the operational airport measurements (Direccion Mete-
195 orologica de Chile (DMC) Servicios Climaticos , 2022) about 1.5 km to the south, reveals good agreement with differences within $\pm 0.5$ K. An exception from this is summer when temperatures rise above about 22°C and our $T_a$ is on average 0.5 K higher. This excess can be explained by the lack of an active ventilation in our measurements. Dew point calculated from our instruments is systematically 0.8 K lower than from the airport data with scatter in the order of 0.3 K. This is equivalent to by 3% too low readings in our relative humidity measurements. Although this is within the range of the instruments accuracy, we
corrected this difference.

### 2.2.5 Sea surface temperature

We use two global Sea Surface Temperature (SST) datasets (GHRSST, 2008, 2018) both with 0.1° and one day resolution to investigate the influence of the ocean. They are based on merged observations by different instruments on different satellites. For times until January 15, 2019 GHRSST (2008) and for times after January 15, 2019 GHRSST (2018) have been used. The
205 datasets overlap between October 2018 and January 2019 and values in the region around Iquique show no differences during this time. From these datasets we extract two single pixel time series 5 km west of the Airport in the ocean (20.5°S, 70.2°W) and 50 km to the southwest in the open ocean (21.0°S, 70.7°W) (Fig. 2).

## 2.3 Synergistic Retrievals

### 2.3.1 Cloudnet

To gain more information about the observed clouds, we have applied the Cloudnet algorithm (Illingworth et al., 2007) which uses zenith observations only of the three remote sensors described above. This algorithm classifies the observed clouds into liquid, mixed phase or ice clouds as well as into a precipitating/non-precipitating class including drizzle detection. The algorithm is based on reflectivity from the CR, brightness temperatures from the MWR and backscatter profiles from the DL. Additionally, data from the European Centre for Medium-Range Weather Forecasting Integrated Forecast System (ECMWF
IFS) provides temperature and wind information throughout the whole atmosphere to be able to discriminate hydrometeor phase. The lidar provides mainly the lowest cloud-base, the MWR information about the presence of liquid water, and the radar about cloud-top and higher cloud boundaries. Doppler velocities from the CR allow identification of falling hydromete-

**Table 2.** Number of days when all instruments could provide data to the Cloudnet retrieval during every month and season. Months are indicated by their first letter, $N$ is the number of days with data, $R$ is the data coverage relative to the days in the respective month or season in percent.

| Month | M | A | M | J | J | A | S | O | N | D | J | F |
|---|---|---|---|---|---|---|---|---|---|---|---|---|
| $N$/days | 6 | 25 | 27 | 0 | 23 | 29 | 30 | 30 | 29 | 27 | 17 | 0 |
| $R$/% | 19 | 83 | 87 | 0 | 74 | 94 | 100 | 97 | 97 | 87 | 57 | 0 |

| Season | MAM | JJA | SON | DJF |
|---|---|---|---|---|
| $N$/days | 58 | 52 | 89 | 44 |
| $R$/% | 63 | 57 | 98 | 49 |

ors, e.g. rain or snow. Together with the IFS temperature profiles, it is possible to identify regions of liquid water or ice clouds. For the cloud frequency of occurrence (FOC) statistics, we focus on warm clouds with bases below 2 km asl. Only clouds with liquid droplets without ice or supercooled droplets are considered. In addition, cases of cloud liquid droplet detection with drizzle are included.

The instruments where equipped with battery backups to ensure controlled shutdown and recover in case of a power shutdown. Nevertheless this did not work in every case for every instrument and lead to several missing days. An instrument failure of the cloud radar lead to no data from end of May to begin of July. And damage of the radar radome forced us to shut down the instrument on January 22, 2019. As the Cloudnet retrieval can only be applied when all instruments provide data this lead to 243 days of data or 72% of the full year. Table 2 shows how many days are per month avaiable. There are few or no days with data in March, June and February and only 57% in January.

### 2.3.2 Boundary Layer Classification

In addition to the standard Cloudnet classification algorithm, we use the boundary layer classification scheme described by Manninen et al. (2018), which is a Cloudnet add-on product. This scheme can identify the regions in the atmospheric boundary layer below the cloud with significant turbulence and determine the origin of this turbulence. The classification is provided as a function of time and height at a resolution of 3 min and 30 m. It is based on calculations of the turbulent dissipation rate from the Doppler lidar vertical velocities, the skewness of the vertical velocity distribution, the derived horizontal wind speeds and backscatter coefficients as well as the near-surface temperature gradient around 2 m height.

At every height and at every time, six classes of turbulence origin are provided: in cloud (if backscatter at a certain height is above a certain threshold), non-turbulent (turbulent dissipation rate below threshold), cloud-driven (skewness of vertical velocity distribution is negative), convective (unstable stratification at surface and turbulent conditions between the considered height and the surface), wind-shear (wind shear above a certain threshold). If a turbulent layer is neither 'cloud-driven', nor 'convective' nor 'wind-shear' it is assigned 'intermittent'. Additionally, the scheme provides information whether the surface layer is stable or unstable.

## 3    Results

### 3.1    Satellite View of a typical Situation

A typical situation off the west coast of northern Chile is depicted in Fig. 2. The ocean on the left side of the scenes is nearly completely covered by stratocumulus clouds while the Atacama, the Andes and the Altiplano to the right are nearly completely
cloud free as they are above 1000 m above sea level and thus higher than the MBL.

The whole cloud deck is typically moving along the coast to the north, turning gradually to the northwest as it approaches the Peruvian coast. This movement can be depicted from the two open cell areas above the ocean which appear at different locations in the morning and noon image. From the displacement we can estimate for these scenes a velocity of around 3 m/s. Typical values go up to 7.5 m/s.

In the morning scene, clouds reach inland at some places: to the northwest of Arica, around Antofagasta, especially at the Mejillones Peninsula at Antofagasta (ANF), and at the southern edge of the scene. All these locations lie at lower altitudes and the edge of the stratocumulus identifies where the landscape reaches cloud-top height. At the coast, in the morning, there are some gaps in the cloud deck, some of which can be related to valleys cutting through the coastal mountain range. At these places dry desert air of the nocturnal flow from the Andes can probably reach the ocean and evaporate the clouds (see e.g.
Schween et al., 2020). Additionally, to southeast of Iquique, a fog field can be seen inland in the central valley between the coastal mountain range and the slopes of the Andes (compare to  Cereceda et al., 2008b; del Rio et al., 2021a; Böhm et al., 2021).

At noon the situation has changed: at some places between Antofagasta and Iquique, larger gaps in the stratocumulus cloud deck opened and reach several tens of kilometers out over the ocean. This typical pattern can be observed nearly every day:
between morning and noon these gaps form at the coast and during the day extend further over the ocean. In the afternoon, clouds form again at the coastal cliff and grow over the ocean. As a result the gaps in the cloud deck are closed in the evening or early night and a continuous stratocumulus cloud field extending from the open ocean to the coast has been reestablished. Some sample plots and a discussion of this winter day and a summer day can be found in the appendix.

### 3.2    Wind

Over the seasons, average wind profiles at Iquique show a clear diurnal course (Fig. 3) with overall low wind speeds during night and higher values during daytime with highest values in the afternoon below 200 m above the surface. During night, wind directions below 500 m are around southeast with speeds of about 2 m/s. Above 500 m one may observe northwest winds. After sunrise, wind direction in the lowest 500 m turn to southwest and gradually increase to reach highest average speeds of 7.5 m/s in the late afternoon (20UTC = 2h before sunset). At higher levels wind direction turns to the left, reaching south in the middle
and east at the top of the observable layer. Wind speeds decrease in the evening and reach night time values again in the first half of the night.

Only a few details vary over the seasons. The range where a retrieval is possible is higher in summer and autumn and low in winter. This is related to the presence of clouds mainly in winter and spring limiting retrieval height during these seasons.

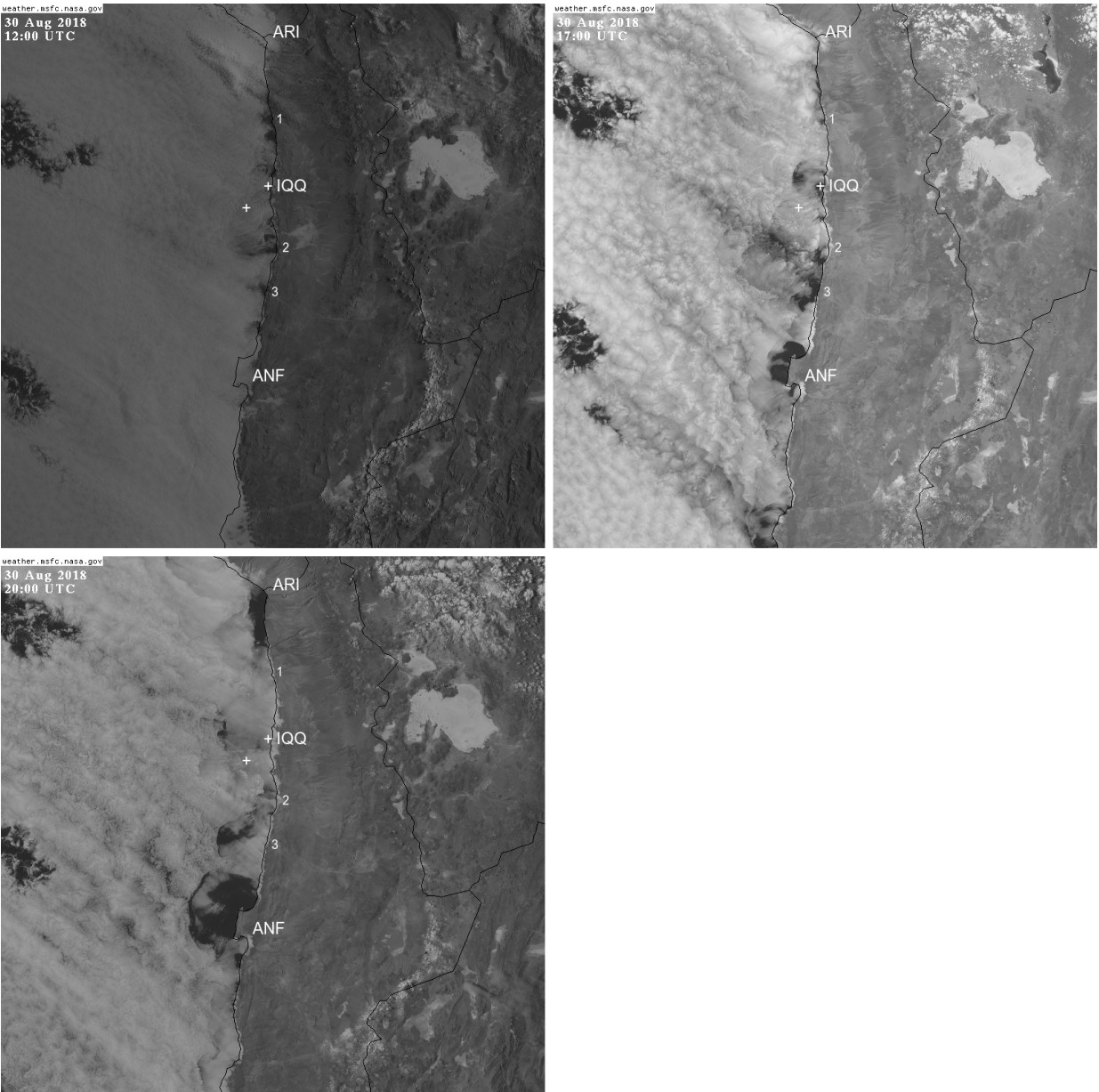

**Figure 2.** Scenes of visible GOES16 channel of August 30, 2018 morning (12UTC, top left) noon (17UTC, top right) and afternoon (20UTC, bottom left). The scenes are centered 70° West and 21.9° South and cover about 800km×800km. Three letter codes indicate the location of the airports of Arica (ARI), Iquique (IQQ) and Antofagasta (ANF). Plus signs indicate positions for which SST was extracted from GHRSST data (GHRSST, 2008). Digits 1-3 indicate locations of valleys cutting through the coastal mountain range. Thin black lines identify the coast, or state boundaries, respectively. The large white area in upper right corner of the satellite scenes is the salt pan Salar de Uyuni. The satellite scenes are courtesy of NASA Marshall space flight center (https://weather.msfc.nasa.gov/goes/). Bottom right shows the Cloudnet classification from Iquique for the same day.

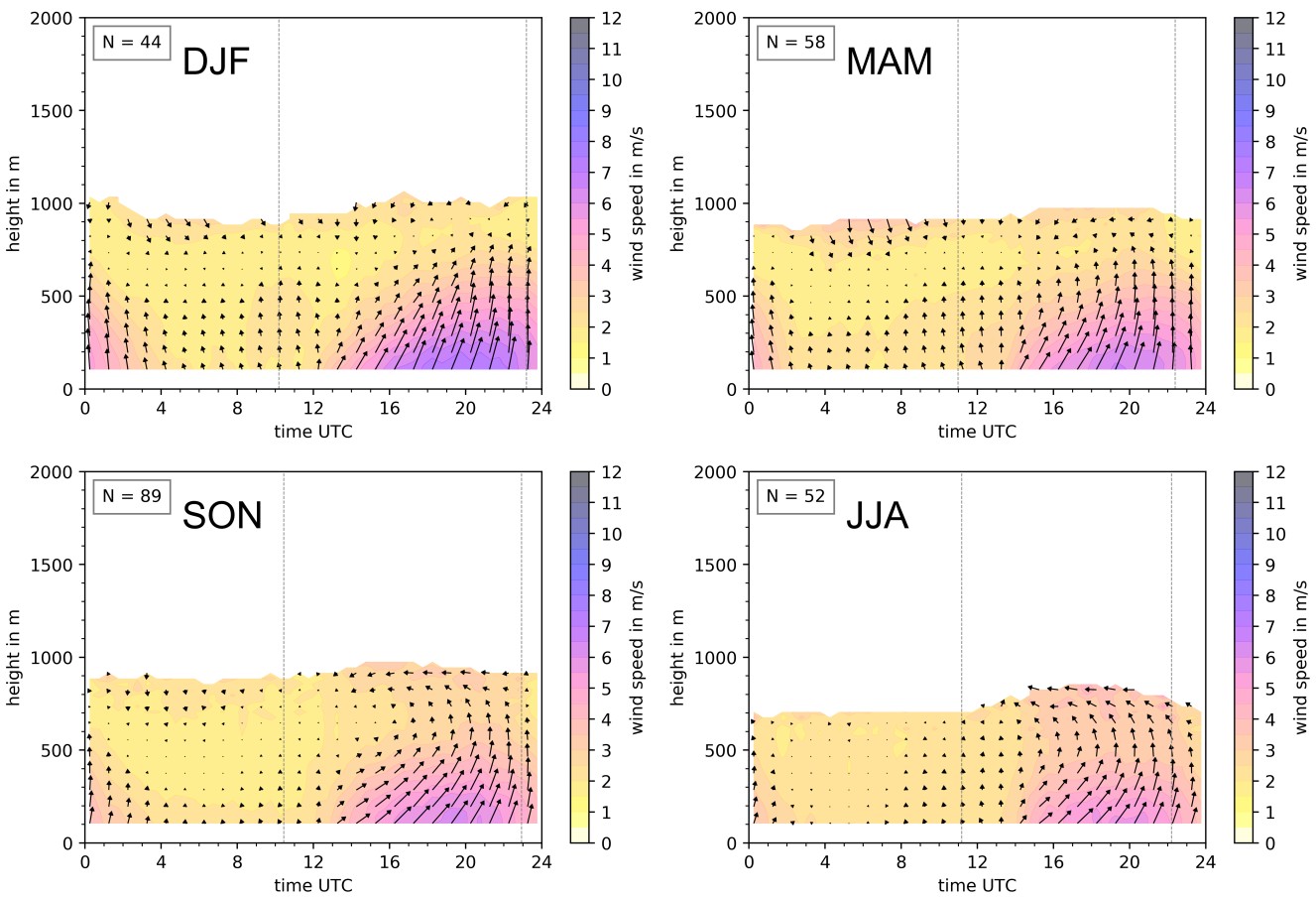

**Figure 3.** Mean horizontal wind speed (shading, length of arrows) and wind direction (arrows) during different seasons as a function of time of day and height. Clockwise from top left: austral summer (DJF), fall (MAM), winter (JJA) and spring (SON). Value of $N$ gives the number of days analyzed. Vertical dashed lines indicate average times of sunrise and sunset. Night appears on the left side, and day on the right side of each plot.

In summer, afternoon average speeds in the lowest 200 m reach 7.5 m/s but only 6.5 m/s or less during the other seasons. This can be related to two things: summer brings more insolation and, as we will see below, there are much less clouds in summer. Additionally, in summer, the center of the southeast Pacific high pressure system lies at the same latitude as Iquique and thus brings stronger winds.

The observed wind pattern resembles a land-sea-breeze: during daytime a sea breeze is present with inflow from the ocean to the land in the lower part of the MBL, and a back circulation towards the ocean in the upper half. The back circulation is stronger in winter and spring when more clouds are present, while the sea breeze is stronger in summer when insolation leads to a stronger temperature difference between land and sea. Superimposed on this is the southerly wind of the Southeast Pacific high pressure system and the South American coastal jet (Muñoz and Garreaud, 2005) which due to channeling along the coastal cliff further increase daytime wind speeds. The land breeze during night is much weaker because the temperature difference between cold ocean waters and the desert is small.

## 3.3 Temperature Profile

We use the Temperature profiles from the microwave radiometer to calculate the potential Temperature with respect to the surface as $\theta = T_{\mathrm{air}} + \Gamma \cdot z$ with $\Gamma = g/c_P$. This is not exact but deviations are below 900 m typically well below 0.1 K. Average potential temperature $\theta$ shows some variation over seasons and time of the day (Fig. 4). In the middle of the maritime boundary layer (MBL) we see average potential temperatures between 15°C (winter) and 21°C (summer), respectively. In the free troposphere at 1.5 km height average potential temperature remains nearly constant at around 32°C over the year. Nevertheless, there are some important features. The MBL is topped by a very strong inversion. Winter data from the radiosonde of Antofagasta, 320 km to the south frequently show a temperature increase of around 10K with maxima up to 17K within less than one hundred meters. Although the MWR cannot resolve these extreme gradients, it clearly shows the inversion with a strong increase of $\theta$ with height and we may regard the maximum of the gradient in this 'radiometer inversion' as the true height of the MBL (see Section 2.2.2). It shows nearly no variation over the year, and is lowest in winter at 950 m and around 1050 m during the other seasons. Despite the coarse resolution of the MWR this agrees with the annual course of the median inversion base height for the period from 1979-2007 in Antofagasta shown by Muñoz et al. (2011). Nevertheless we have to consider that cloud base, and thus probably also inversion height in Iquique, is on average by about 180 m higher than in Antofagasta (Muñoz et al., 2016). Additionally the inversion height data from the Antofagasta radiosonde (Muñoz et al., 2016) show beside a weak decrease of $-16$ m per decade, a large year to year variability in the order of $\pm 100$ m. This makes it difficult to put our observations in relation to these long term analyses.

Over all seasons MBL height shows no diurnal variation (Fig.4). Similar temperatures above the inversion show no significant variation over the year.

In summer and autumn, close to the surface, values of $\theta$ are typically lower than at the bottom of the inversion, i.e. stratification is stable in contrast to winter and spring when it is neutral. This has strong implications for the presence of clouds as a stable stratification inhibits supply of water to the clouds. Lobos-Roco et al. (2018) have shown that during stable stratification

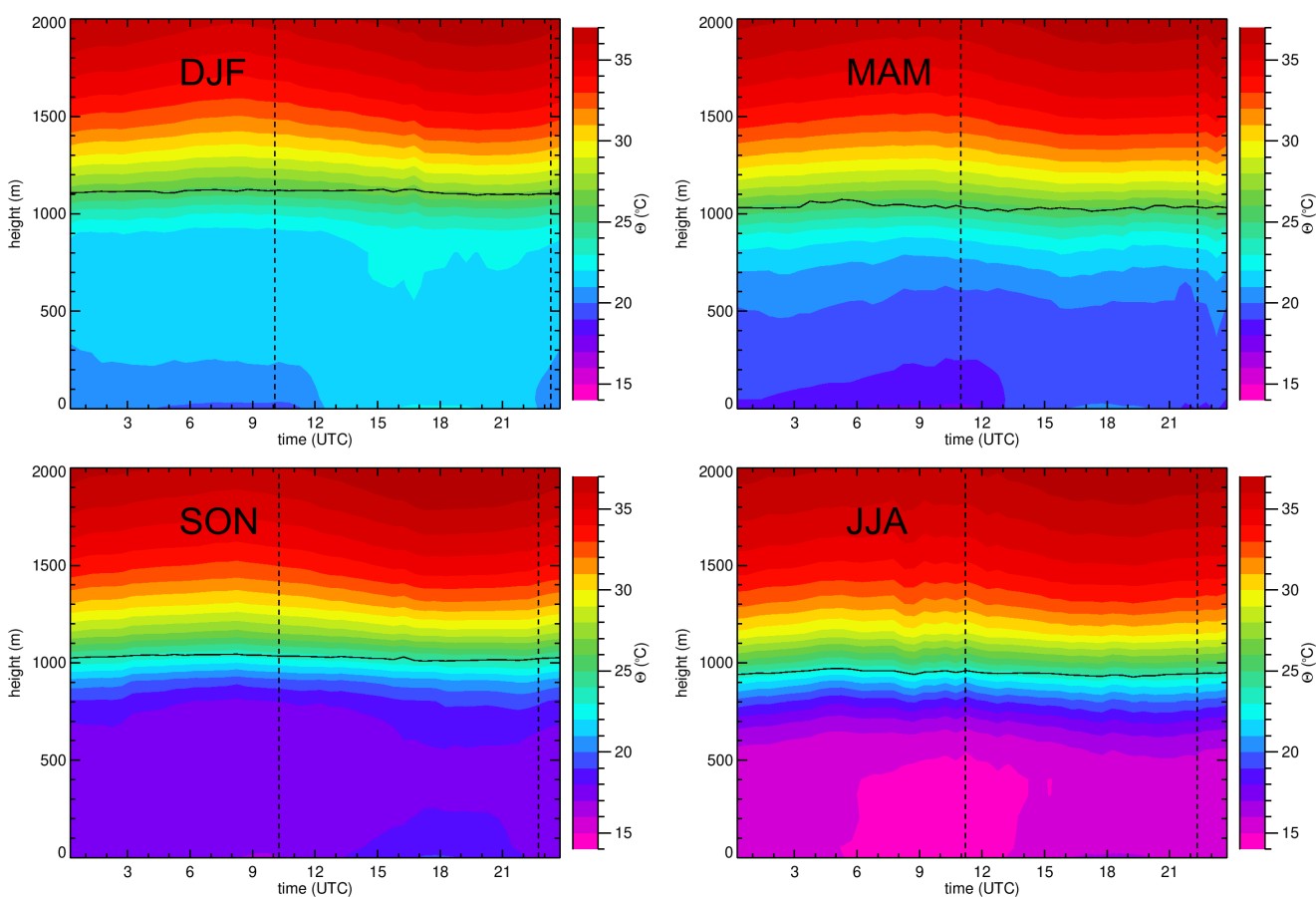

**Figure 4.** Mean potential temperature during different seasons as a function of time of day and height. Clockwise from top left: austral summer (DJF), fall (MAM), winter (JJA) and spring (SON). Color shading is based on three month averages of 1 hour and 30 m intervals. Black line depicts the average height of the strongest gradient. Vertical dashed lines indicate average times of sunrise and sunset. The analysis here is restricted to the same days as the Cloudnet retrieval.

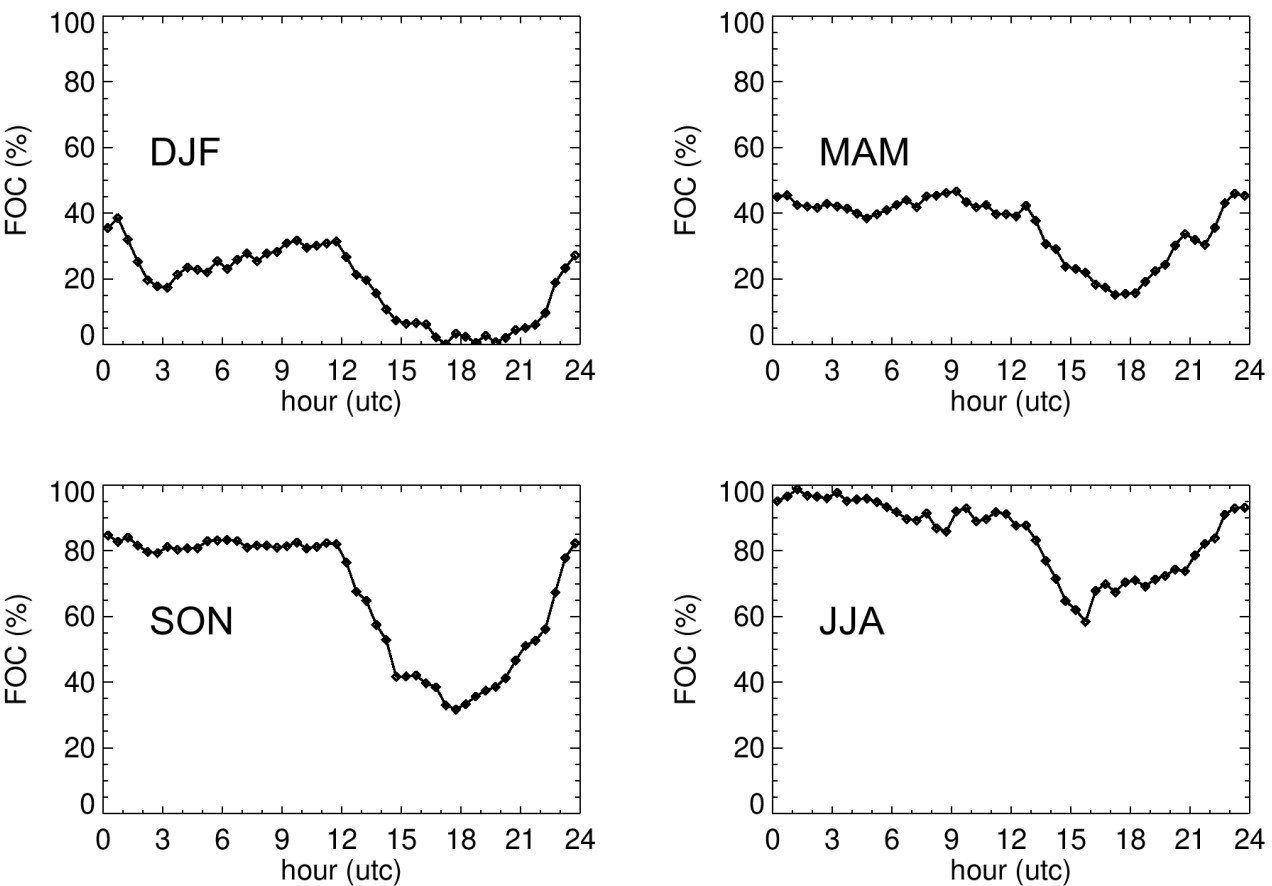

**Figure 5.** Diurnal course of frequency of occurence of clouds detected by Cloudnet with cloud-base <2 km asl during different seasons. Clockwise from top left: austral summer (DJF), fall (MAM), winter (JJA) and spring (SON). Value of $N$ gives the number of days analyzed. Vertical dashed lines indicate average times of sunrise and sunset.

between their site at 1100 m ASL and the airport at 53 m, typically no clouds are present. Accordingly we can expect that during the summer months less clouds will be present.

### 3.4 Cloud Occurrence

The Cloudnet target classification scheme is used to investigate the frequency of occurrence (FOC) of warm clouds with cloud-base lower than 2 km asl (Fig. 5). Most clouds occur in winter and spring, when during night about 90% of the time clouds are present, while during summer and autumn this reduces to 25% and 45%, respectively. During daytime FOC reduces by about 30%-points compared to the night meaning nearly no clouds during summer afternoon and somewhat more clouds in autumn and spring. In winter FOC stays allways above 60%.

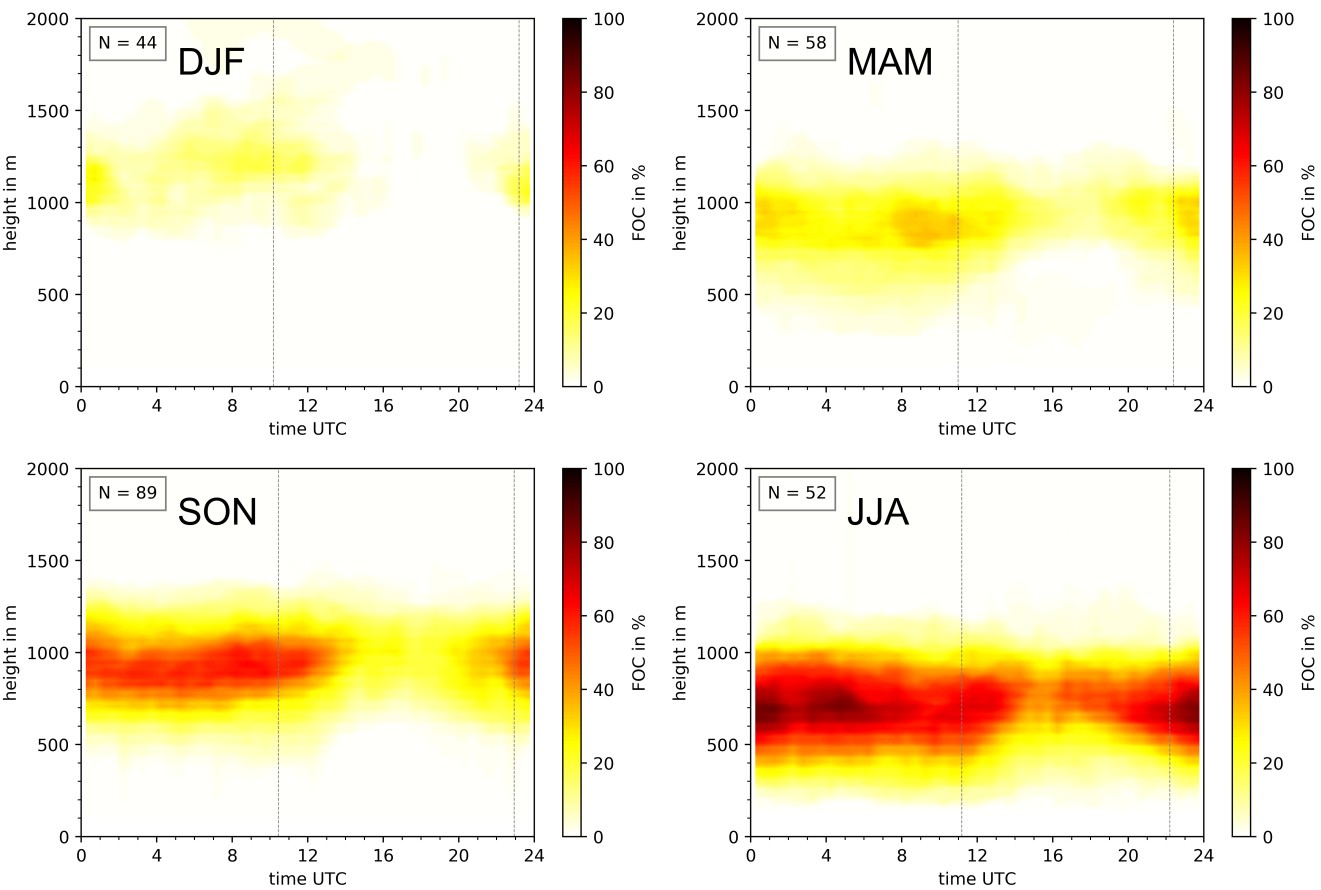

**Figure 6.** Frequency of occurrence (FOC) of clouds with cloud-base below 2 km asl in time×height bins during different seasons. Clockwise from top left: austral summer (DJF), fall (MAM), winter (JJA) and spring (SON). Value of $N$ gives the number of days analyzed. Vertical dashed lines indicate average times of sunrise and sunset.

The vertical structure of the cloud varies as well (Fig. 6). In general clouds can be found at higher levels in summer than in the other seasons with lowest clouds in winter. But while in summer, and to some extent in autumn, there is no clear cluster where we can expect clouds, during winter and spring they are clearly confined to a height range from 300 m to 1100 m in winter and 600 m to 1200 m in spring. The diurnal course reveals that the reduction of cloud occurrence between noon and afternoon occurs from below. The lower boundary of the region where to expect clouds, increases from morning hours to the

afternoon by several hundreds of meters. This is in agreement with the observations of del Rio et al. (2021b) made some km to the south-east at the slopes of the coastal mountains in another year.

    The same pattern can be observed if we investigate average cloud boundaries (Fig. 7). Over the whole day average cloud-top is at about the same height: 1000 m in autumn, 1100 m in spring and 900 m in winter. In contrast hereto, cloud-base is lowest during night and increases from sunrise until afternoon by about 50 m (spring) to 100 m (autumn and winter). An exception

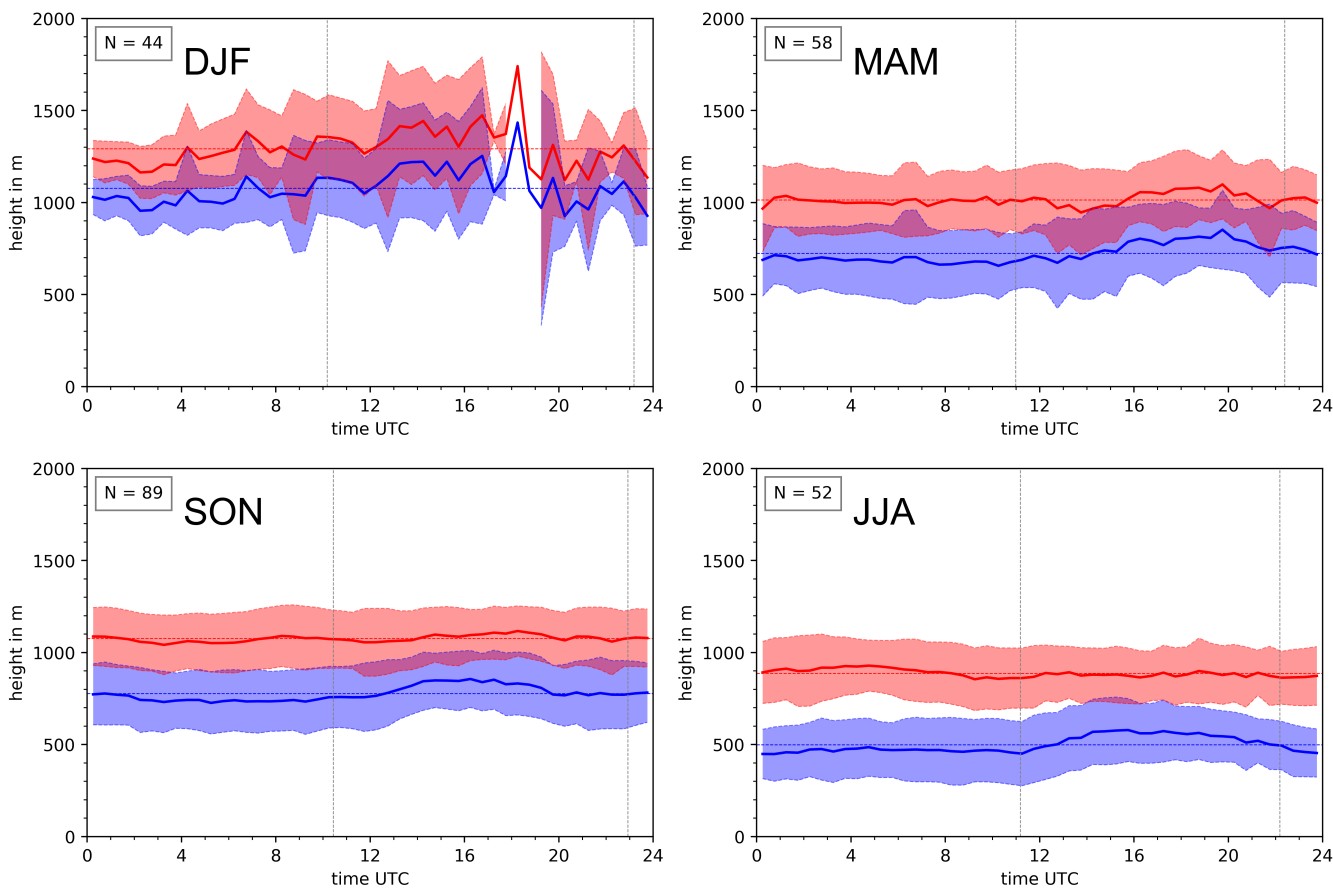

**Figure 7.** Average diurnal course of cloud-base (blue) and cloud-top (red) during different seasons. Clockwise from top left: austral summer (DJF), fall (MAM), winter (JJA) and spring (SON). Color shading indicates the range of one standard deviation. Value of $N$ gives the number of days analyzed. Vertical dashed lines indicate average times of sunrise and sunset.

from this pattern occurs in summer when cloud-base as well as cloud thickness increases over the day which is more typical for shallow cumulus clouds. Nevertheless, cloud occurrence in summer is low and the variability of the cloud boundaries is larger than cloud thickness, such that these numbers should be interpreted carefully. Nighttime cloud-bases are highest in summer (around 1100 m), intermediate in autumn and spring (700 m-800 m) and lowest in winter (500 m). Cloud thickness is about 300 m during autumn and spring and 400 m in winter nights. Muñoz et al. (2016) found in a 33 year dataset from Iquqique, centered around 1997, a similar pattern with cloud bases higher during daytime than during night. Nevertheless they found lower night to day amplitudes (50 m-75 m) and also cloud bases which were especially in autumn and winter by about 200 m-300 m higher then we observed now. This difference can only partly explained by the trend of -100 m/15yr they found. From this comparison it seems as if this trend has accelerated.

The diurnal course with a constant cloud-top and a rising cloud-base during daytime indicates that processes within the MBL are the reason for the observed thinning of the cloud deck during daytime. A possible mechanism to explain this could be the absorption of solar radiation during daytime, so that long-wave radiative cooling at cloud-top is largely cancelled out and the formation of subsiding turbulent eddies at cloud-top is inhibited. As a result the sub-cloud layer is not well-mixed, vertical moisture transport from the ocean is at least reduced, water loss by detrainment and thus evaporation into the free troposhere at cloud-top is not compensated anymore and the MBL becomes drier. A lower water vapor content means a higher cloud-base or even no cloud (Bretherton et al., 2004). However, this cannot explain why the cloud deck starts to evaporate at the coast. Another mechanism could be the general circulation pattern at the coast in which air flows over land. While flowing over warm solid grounds air heats up which increases the lifting condensation level. This could explain why clouds evaporate especially at the coast.

## 3.5 Liquid Water Path

As discussed above (sect. 2.2.2) we use four different retrievals based on different pre-processing of the radiosonde data to derive the liquid water path (LWP). The average difference between the resulting values is small (3 g/m$^2$) but shows a diurnal course with larger values during night (up to 18 g/m$^2$). The theoretical retrieval uncertainty of LWP is determined to 10 g/m$^2$. Calibration and absorption model uncertainties can lead to systematic LWP offsets which can mount up to a maximum error of 20-30 g/m$^2$. These effects have been corrected for by applying a clear-sky offset correction, which is applied to all-sky observations. Thus, we expect a random uncertainty for individual values of not significantly more than 10 g/m$^2$, which further reduces for seasonal averages by a factor of $1/\sqrt{N}$. I.e. LWP uncertainty is dominated by the uncertainty of the radiosonde data preprocessing.

In general, average LWP is lowest in summer (average value 20 g/m$^2$) and highest in winter (70 g/m$^2$) (Figs. 8 and 9). Diurnal courses show a recurring pattern with maximum values during night and a minimum in the early afternoon. Daily amplitudes are largest in winter (70 g/m$^2$), smaller in autumn and spring ( 25 g/m$^2$ and 50 g/m$^2$) and nearly negligible in summer (17 g/m$^2$). The diurnal amplitude in winter and spring equals the average values, i.e. the clouds loose half of their water on average during daytime. Standard deviation of the half hourly averages of the diurnal courses, i.e. day-to-day variability for the respective hours is in the order of the average values itself (not shown). It is noteworthy that in winter (JJA) the maximum LWP is found in the first half of the night and values decrease thereafter. This might be explained by occurrence of drizzle especially in the nights of August when during up to 40% of the time drizzle occurs. Drizzle removes liquid water from the cloud and eventually evaporates and thus removes liquid water from the cloud. While night time average values in spring (60 g/m$^2$) are significantly lower than those reported by Bretherton et al. (2004) for the open ocean in October (100-220 g/m$^2$), they agree well with those reported for the Chilean coast (Mechoso et al., 2014).

## 3.6 Drizzle

Drizzle plays an important role in the moisture budget of stratocumulus clouds (Wood, 2012) and it may also be an additional water supply to the coastal mountains and its fog oases. We therefore investigate here how often drizzle occurs and how far

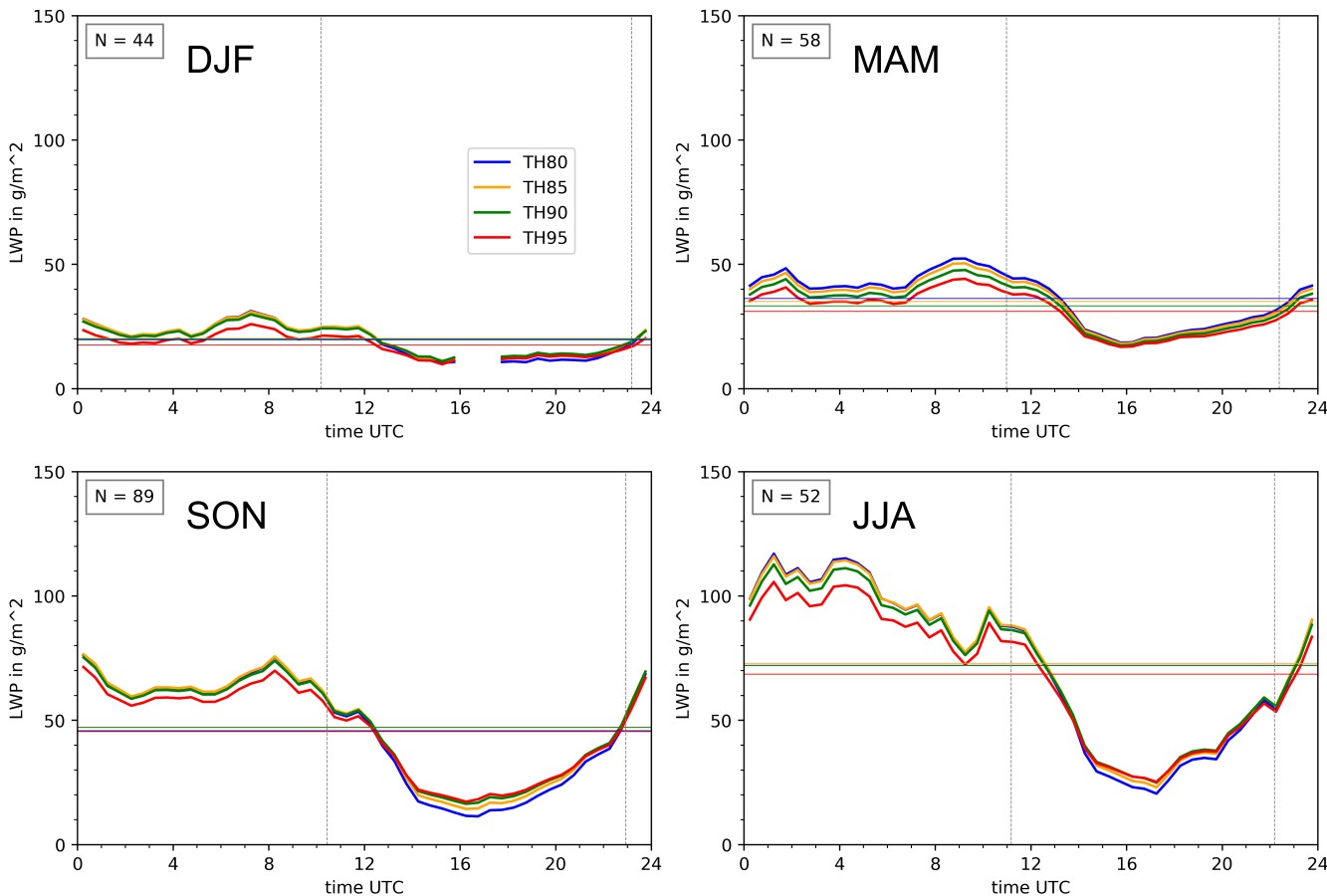

**Figure 8.** Average diurnal course of liquid water path (LWP) during different seasons. Clockwise from top left: austral summer (DJF), fall (MAM), winter (JJA) and spring (SON). Different colors depict different LWP retrievals (see sect. 2.2.2). Horizontal lines indicate seasonal averages. Value of $N$ gives the number of days analyzed. Vertical dashed lines indicate average times of sunrise and sunset. Gap around 17UTC in austral summer (DJF) is due to times when the sun is in zenith making retrieval impossible.

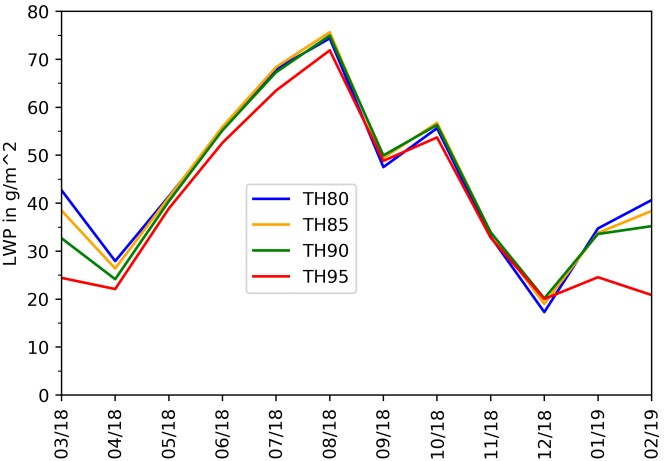

**Figure 9.** Annual course of monthly means of liquid water path (LWP). Different colors depict different LWP retrievals (see sect. 2.2.2).

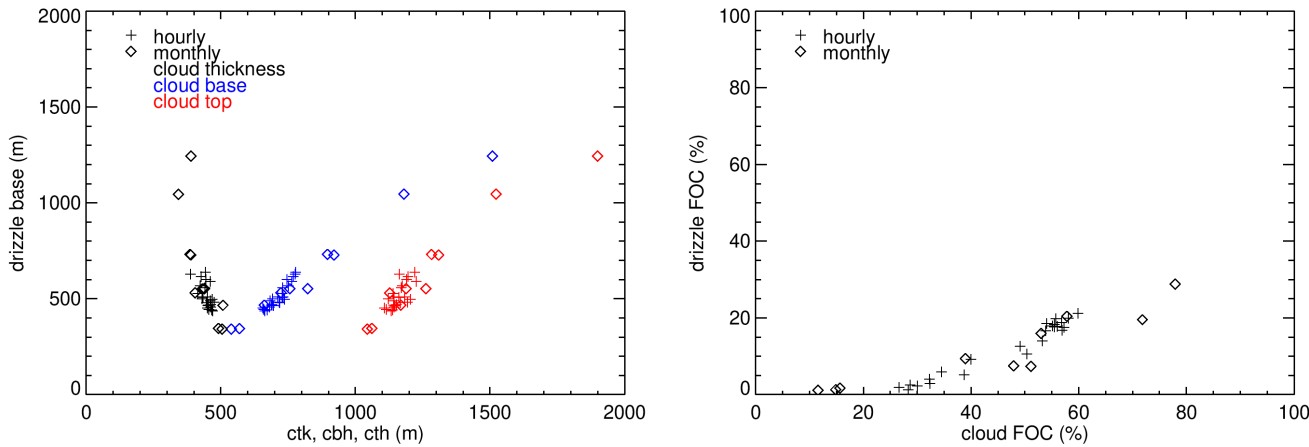

**Figure 10.** Left: drizzle basis (height at which drizzle evaporates completely $z_{de}$) as a function of cloud-base (blue), cloud-top (red) and cloud thickness (black). Right: Frequency of occurrence (FOC) of drizzle at any height as a function of FOC for clouds. Monthly averages over all hours are denoted by a +-symbol, hourly averages over all months for different hours are indicated by a ◇-symbol.

down it reaches. The Cloudnet classification scheme provides two classes "*drizzle or rain*" and "*drizzle or rain coexisting with cloud liquid droplets.*" As the weather station never detected rain we assume that these classes always identify drizzle and we can investigate at which times and heights drizzle occurs below and within the stratocumulus. We consider all cases with drizzle and a cloud present below 2 km, and count all cases with drizzle in every hour over the whole year, and additionally in every month over all hours. We use these numbers to calculate the relative amount of time or frequency of occurence of drizzle per month and per hour. We also identify the lowest level to which drizzle reaches. During the whole observation period, drizzle evaporated on average at 485 m and reached only sporadically to the lowest Cloudnet level of 200 m. Regarding monthly averages, the lowest average height where drizzle occurred was in June and July at 300 m above ground, while on a hourly basis over the whole year the lowest average drizzle base was 450 m during night. Nevertheless, it should be noted that drizzle may reach the ground further inland where altitudes increase fast with distance to the coast. The height $z_{de}$ at which drizzle evaporated completely, showed some dependence on cloud-base and cloud-top: the higher cloud-base and top, the higher is $z_{de}$ with thicker clouds leading to lower $z_{de}$ (Fig. 10, left panel).

The frequency of occurrence of drizzle follows that of clouds with a difference of 30-40%-points (Fig. 10, right panel). This is observed in the monthly averages as well in the diurnal course where most of the drizzle occurs during night with on average during 20% of the time reaching up to 47% in the nights of August (not shown).

### 3.7 Turbulence

Mean standard deviation of vertical velocity ($\sigma_w$, Fig.11) remains most of the time below 0.5 m/s. Only during daytime higher values develop from the ground and reach up to 300 m in summer and autumn and 750 m in winter and spring. During night time the upper half of the MBL shows especially in winter and spring slightly higher values.

The mean diurnal course of the skewness (Fig.12) shows distinct night and day patterns: During night skewness is mostly negative with lowest values in the upper half of the MBL, while during daytime skewness in the lower 2/3 of the MBL is clearly positive. Nighttime skewness is most negative in winter and spring. Daytime positive skewness reaches higher up in winter and spring than in summer and autumn. Most positive skewness occurs at heights in the range 200-300 m.

Vertical velocity in rising, as well as subsiding plumes is typically concentrated in narrow chimneys with high speeds surrounded by larger regions with only small velocities compensating for the mass transport in the plume cores. As a result regions with rising plumes show positive skewness and regions with subsiding plumes show negative skewness. Accordingly night time turbulence is driven mainly by subsiding plumes, and day time turbulence is driven by plumes rising from the surface. Nighttime as well as day time turbulence in winter and spring is more intense and reaches further down and further up, respectively, when compared to the other half of the year.

To further investigate how frequent which type of turbulence occurs and how far its influence reaches we use the boundary layer classification scheme of Manninen et al. (2018). We are mainly interested whether there is turbulence and whether this turbulence is connected to the cloud ('cloud-driven'), the surface ('convective'), wind shear, or whether its source cannot be identified ('intermittent'). Whereas in summer and autumn a significant amount of the time (45-65%) is classified as non-turbulent, winter and spring show much less calm moments (5-35%, see Fig. 13). Convective turbulence is present during

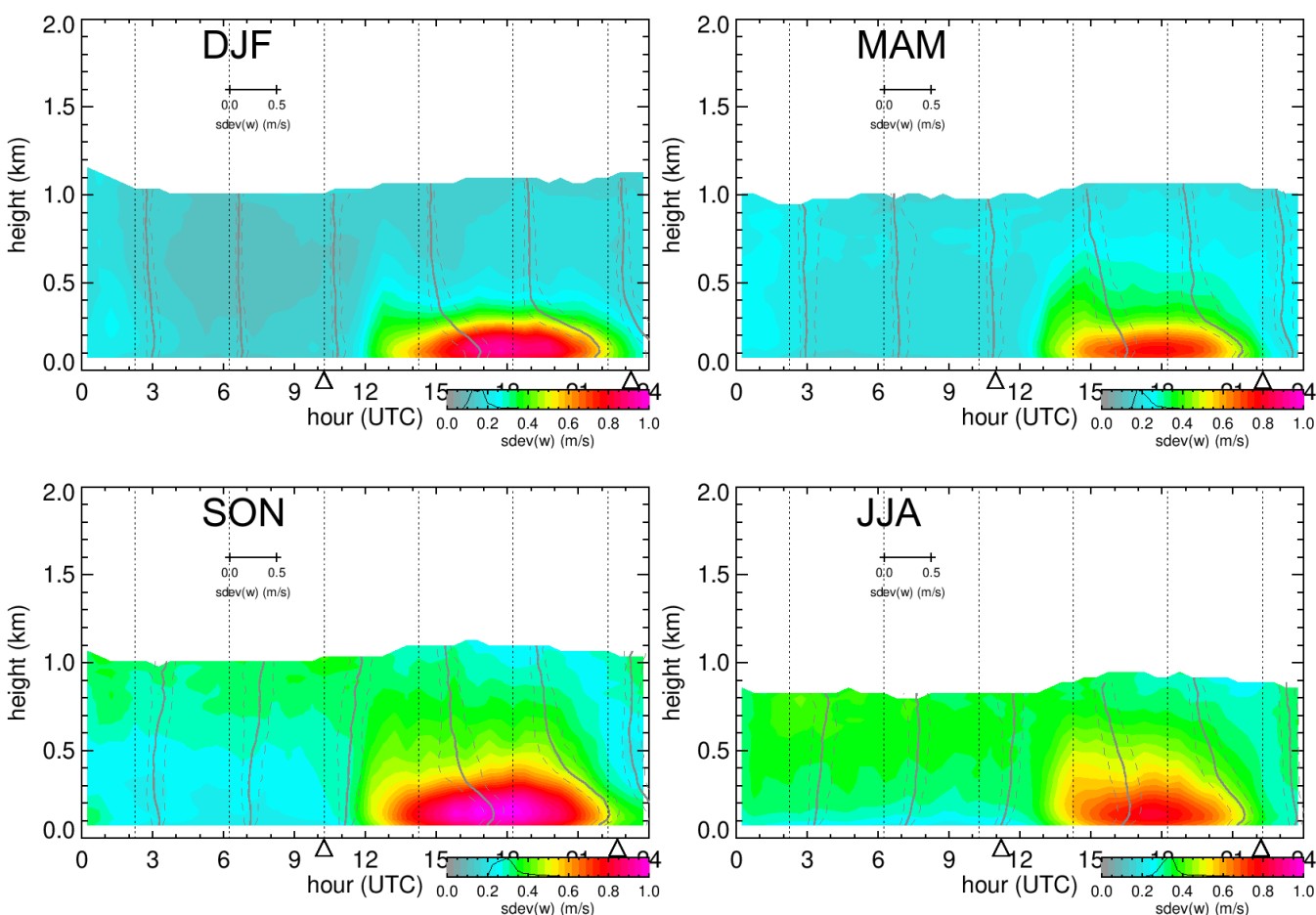

**Figure 11.** Time-height sections of mean standard deviation of vertical velocity (color shading, calculated as described in Schween et al., 2014), for, clockwise from top left, summer, autumn, winter, spring. Gray lines are the median profiles (solid) and interquartile range (dashed) at the times indicated by the dotted vertical lines. Time is given in hours UTC such that night appears on the left side and daytime on the right of each plot. Triangles at the abscissa indicate average times of sunrise and sunset, respectively.

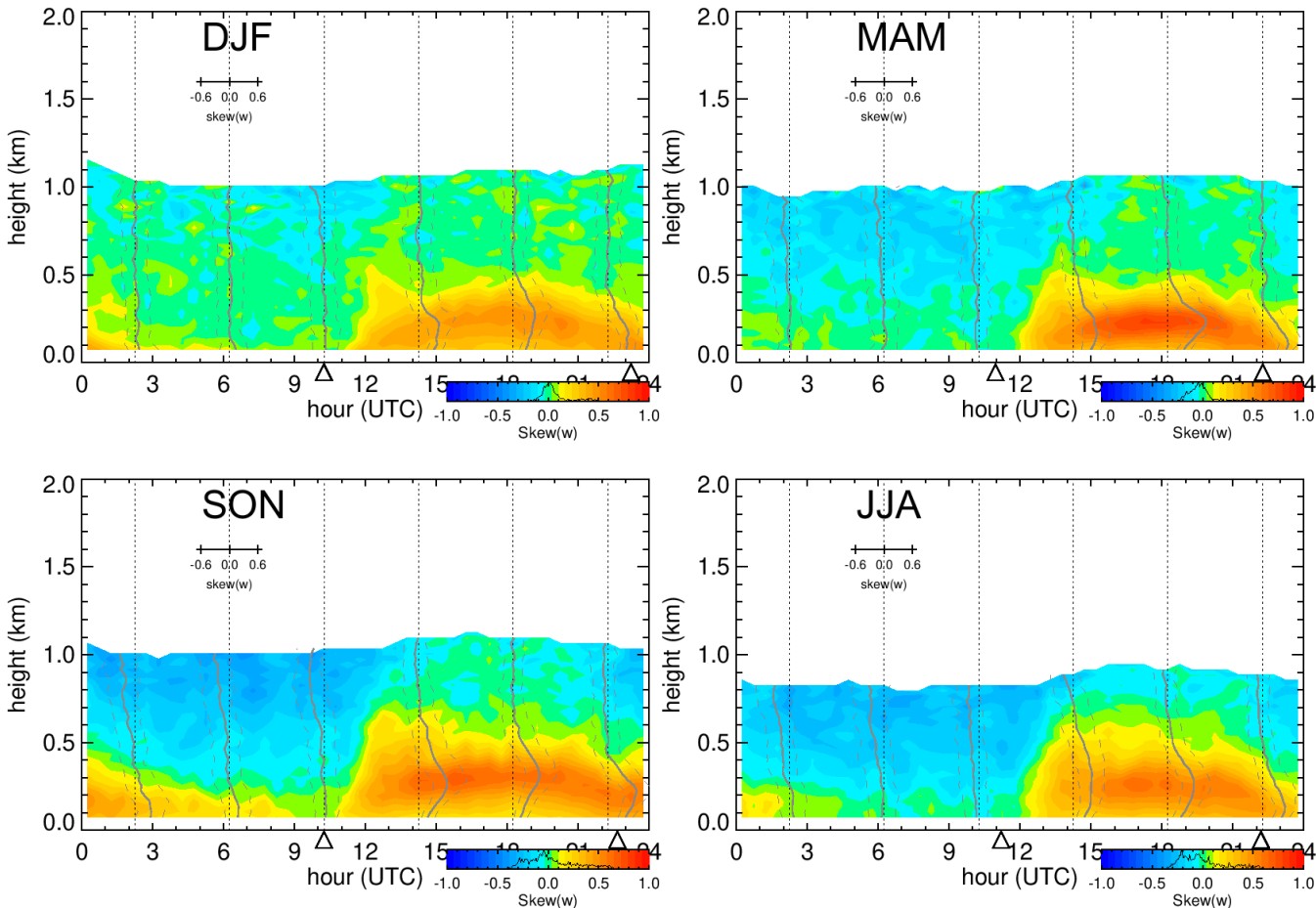

**Figure 12.** Time-height sections of vertical velocity skewness during the four seasons (color shading). Gray lines are the median profiles (solid) and interquartile range (dashed) at the times indicated by the dotted vertical lines. Time is given in hours UTC such that night appears on the left side and daytime on the right of each plot. Triangles at the abscissa indicate average times of sunrise and sunset, respectively.

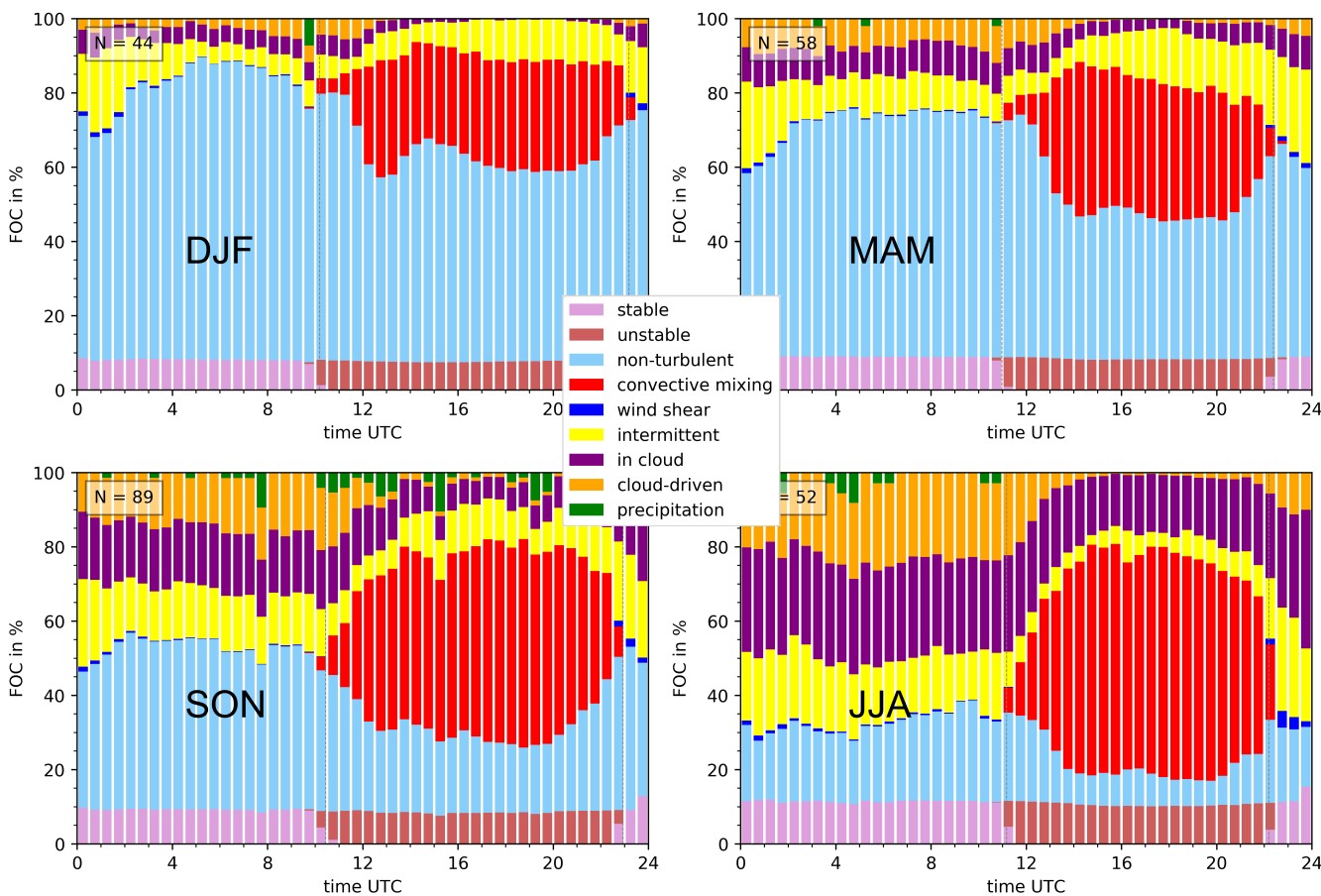

**Figure 13.** Diurnal course of frequency of occurrence (FOC) of boundary layer classes during different seasons. Clockwise from top left: austral summer (DJF), fall (MAM), winter (JJA) and spring (SON). Value of $N$ gives the number of days analyzed.

daytime in all seasons when stratification at the surface is unstable, but most frequently in winter and autumn (up to 65%). The same is valid for cloud-driven turbulence which occurs most frequently in winter nights (20%) while it is nearly not detected in summer and rarely in autumn (8%).

The vertical distribution of turbulence classes reveals that, although rarely occurring, in summer nights cloud-driven turbulent mixing may nearly always reach down to the surface (Fig. 14). In winter nights cloud-driven turbulence occurs much more frequently (30% at average cloud-base height) but its frequency decreases from cloud-base towards the surface by about one third. Given the typically low winds during night, and thus the lack of other possible sources for turbulence, one might speculate that the origin of 'intermittent' turbulence during night is also due to the clouds, although vertical velocity does not show negative skewness. This would increase the frequency of cloud driven turbulence and remove the height dependence.

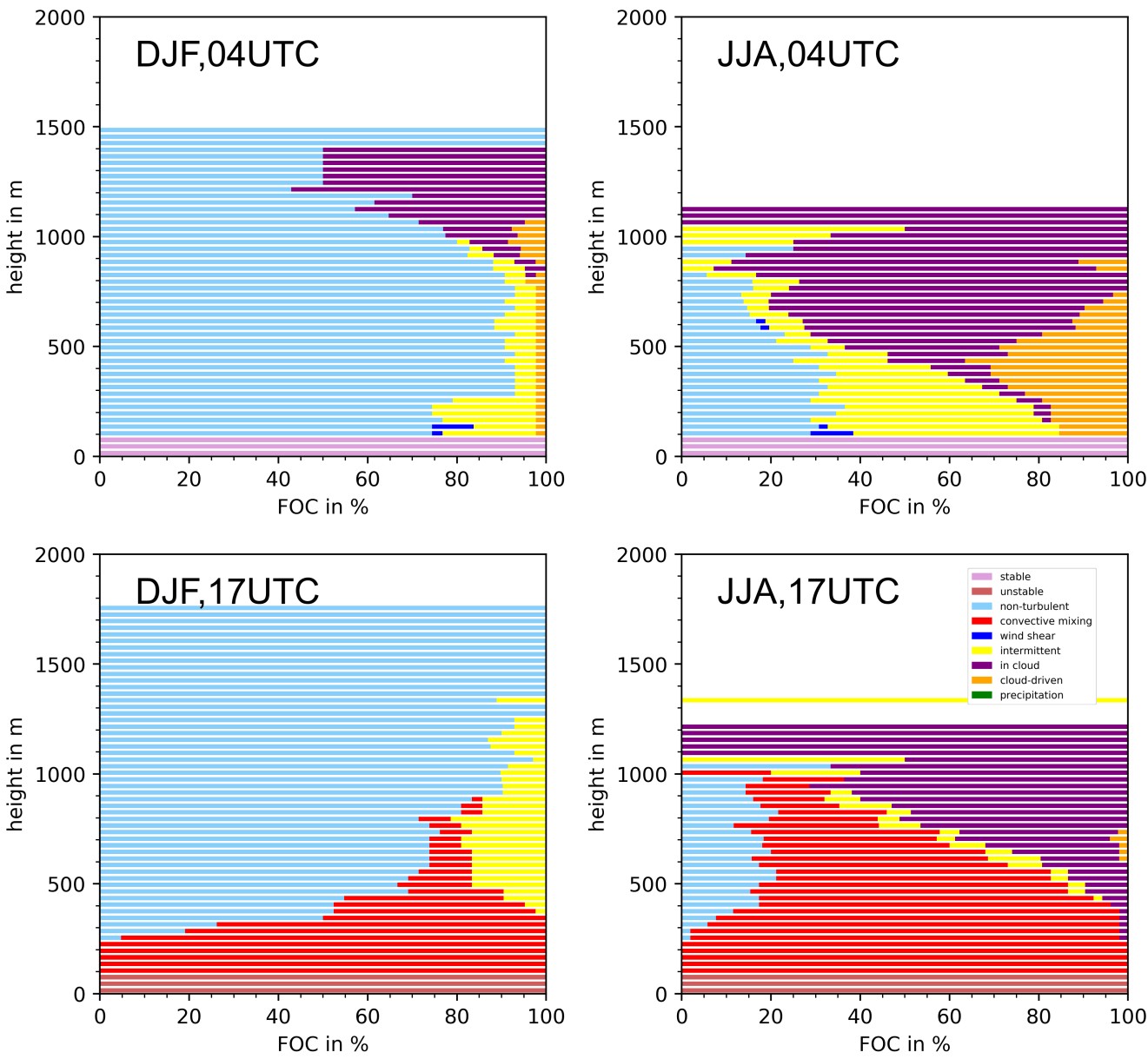

**Figure 14.** Frequency of occurrence of boundary layer classes at different heights in austral summer (DJF, left column) and winter (JJA, right column), around local midnight (4UTC=0LT, top row) and early afternoon (17UTC=13LT, bottom row).

Under this assumption turbulence originating from the clouds reaches the surface in every case and thus mediates the transport of moisture from the ocean into the MBL.

    During daytime at local noon turbulence is mainly convective, i.e. it is driven by heating of the ground. But while in summer it stays mostly confined to the lowest 500 m above the ground and well below the average cloud-base at 1100 m, in winter it can reach up to 1000 m and is typically connected to a cloud. This seasonal difference in convective turbulence reflects the
415 different stratification during the two seasons.

## 4    Discussion

To be able to interpret the observations, we first review the mechanisms which maintain the marine stratocumulus (see e.g. Wood, 2012). Based on this and the observations described so far, we then try to give a holistic picture of the seasonal and daily stratocumulus situation at the northern coast of Chile.

Tropical stratocumulus clouds are typically capped by a strong inversion and a very dry layer above. They continuously loose water by evaporation to the dry, free troposphere. As they exist permanently over large regions, effective mechanisms must exist to balance this water loss. Especially during night, the difference of long-wave radiation emitted by the stratocumulus cloud-top and the low downward emission of the dry atmosphere above, leads to significant radiative cooling at cloud-top. Evaporative and radiative cooling at cloud-top generates pockets of cold air that organize into plumes of cold air that subside towards the
ocean surface. The movement of the descending plumes in the stratocumulus capped MBL is compensated by ascending air, these interchanging up and downward movements generate a well mixed layer. This is similar to a Convective Boundary Layer (CBL), where plumes with positive buoyancy are generated at the warm surface and ascend through the whole boundary layer leading to intensive mixing and neutral stratification. The stratocumulus capped maritime boundary layer (SCBL) thus can be seen as an "upside down convective boundary layer" with similar mechanisms. Comparable to a CBL, intense turbulent mixing
in the SCBL leads to neutral and moist-neutral stratification below the cloud and in the cloud, respectively. And like the CBL which warms up as a whole, the SCBL cools as a whole until it approaches sea surface temperature (SST) at its bottom. And similar to convective plumes, which due to their inertia can reach up into the inversion at CBL top and mix warm air into the CBL, cold air plumes in a SCBL can reach through a sufficient thin stable layer at the sea surface. In this case descending plumes can reach the ocean surface, moisture from the ocean surface is mixed upwards to the cloud and thus can compensate
for the moisture loss to the free troposphere. In contrast hereto a stable stratification over large ranges of the SCBL would inhibit the replenishment of water to the cloud as it inhibits the descent of plumes to the sea surface.

    As long as the SCBL is well mixed and in a stable state, cloud-top energy- and mass-fluxes must be compensated by fluxes at its bottom. If these fluxes are not balanced, the SCBL will change its temperature and/or water content. Radiative emission at cloud top must be balanced by sensible heatflux from the ocean surface. Loss of water vapor by detrainment into the free
troposphere must be compensated by evaporation at the ocean surface. Drizzle removes liquid water from the cloud. But as long as it does not reach the surface it does not change the water content of the MBL. A further factor is large scale subsidence. In a CBL warm rising plumes cross the inversion and thus lead to a growth of BL height and this growth is reduced by subsidence

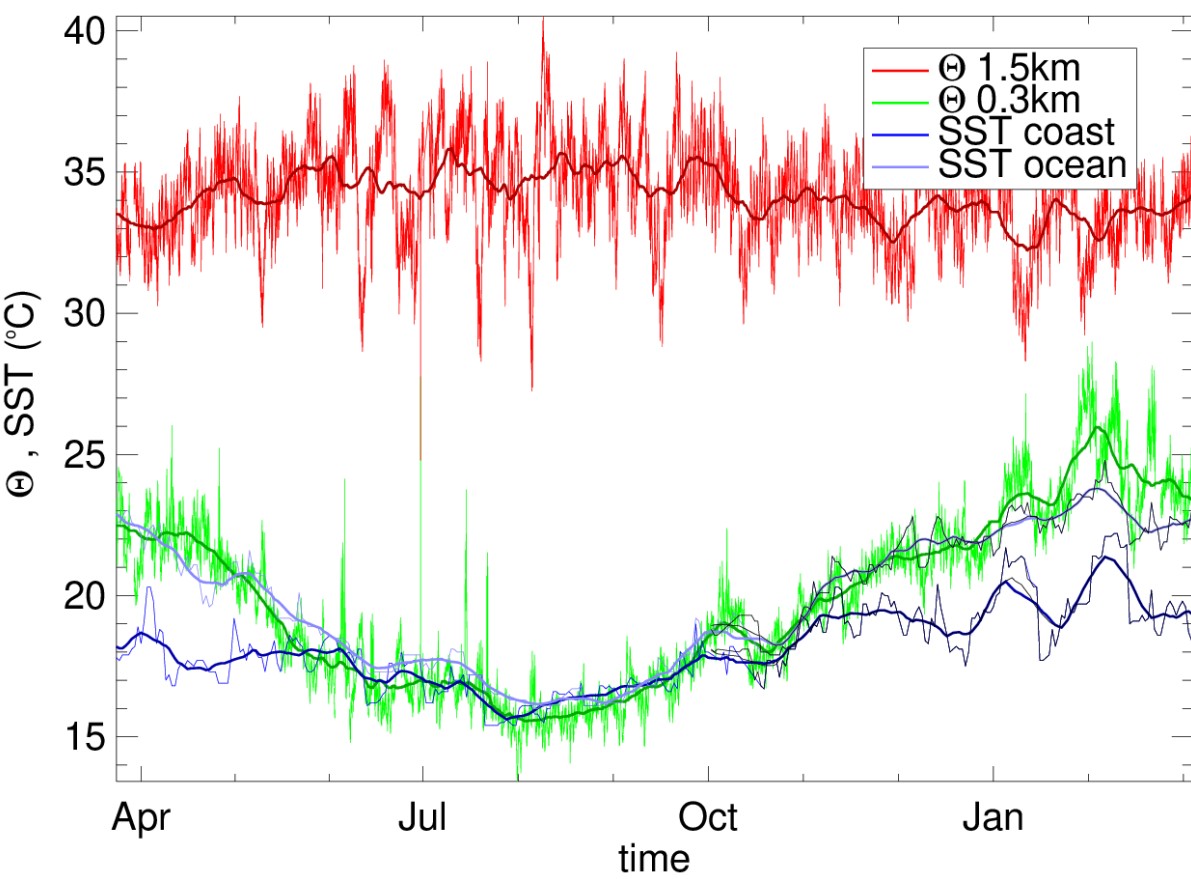

**Figure 15.** Air and ocean temperatures from April 2018 to March 2019 with potential temperature with reference pressure at sea level ($\Theta$) from MWR measurments for 1.5 km (red) and 0.3 km (green) height, sea surface temperature (SST) 5 km from the coast (light blue) and 50 km to the southwest in the open ocean (dark blue). Thin lines indicate 15-minute ($\Theta$) or daily (SST) values, whereas thick lines are 30-day gliding averages.

(see e.g. Ouwersloot and Vila-Guerau de Arellano, 2013). In a SCBL this role of resistance against subsidence is taken by the entrainment of air from above the BL (Wood, 2012), nevertheless entrainment rates remain a factor of uncertainty.

## 4.1  SST and Seasonal Forcing

To investigate the relation of the MBL temperature to SST in our data set, we calculate the potential temperature $\theta_s$ with sea surface as reference level. We use $\theta_s$ at 0.3 km derived from the MWR measurements as a measure for $\theta_s$ in the whole MBL and $\theta_s$ at 1.5 km as a measure for the free troposphere temperature.

We provide SST from two points: one from 5 km west of the Iquique site which is representative for the temperatures of coastal waters, and a second point in the open ocean 50 km to the south-west. The open ocean point has been chosen with the idea that air arriving at Iquique with wind speeds between 2 and 7 m/s from the south-west have passed this point between two, and seven hours before, respectively. The SST close to the coast at Iquique shows an annual cycle with a minimum in August (15.5°C), and maximum in February (21°C, see Fig. 15). The open ocean SST shows a larger amplitude with summer maxima above 25°C whereas in winter it can go down to below 15°C. This larger amplitude is due to variations in the location of the Humboldt Current. As can be inferred from the global SST datasets (GHRSST, 2008, 2018), at some point the Humboldt current turns to the north-west before it reaches the coast of Peru. In summer this turn occurs further to the south and warmer waters propagate from the north along the coast of the continent. Nevertheless, upwelling still leads to a stripe of colder waters direct at the coast of roughly 50 km width.

Our observations show that from June to beginning of November, the MBL at the coast has the same temperature as the ocean surface and is thus in equilibrium with the underlying ocean. During the remaining months between mid of November and May, the MBL is warmer than the coastal ocean and thus decoupled from it. It is instead in equilibrium with the warmer waters 50 km kilometers off the coast and has been advected by the mean wind to the colder coastal waters. Temperature in the free troposphere at 1.5 km height shows only a weak annual variation which is smaller than the day to day variability. As a result the temperature difference between free troposphere and MBL is large in winter and low in summer (Figs. 4 and 15). A strong temperature difference or strong inversion decouples the MBL from the free troposhere and thus inhibits the loss of moisture to the free troposhere.

When the MBL has the same temperature as the ocean, plumes from cloud-top can reach the ocean surface and moisture from the ocean is mixed into the boundary layer and the cloud layer. In winter we have accordingly two mechanisms which support a persistent thick stratocumulus cloud deck: coupling of the MBL to the ocean surface and decoupling of the MBL from the free troposhere by a strong inversion. These winter conditions are accompanied by stronger large-scale subsidence caused by the then closer located center of the south-east pacific high pressure system. This subsidence further sharpens the inversion and slightly lowers its height (Fig. 4) compared to summer. As a result the winter season coincides with times of most frequent cloud presence (sect. 3.4, fig. 6). In contrast, summer conditions are less favourable for a persistent stratocumulus: Large scale subsidence is smaller and accordingly the inversion is less sharp and lies slightly higher. The weaker inversion allows a larger moisture loss to the free troposhere whereas the stable stratification of the coastal MBL reduces the moisture supply from the ocean.

We believe that our one year data show that it is not alone the absolute value of the SST at the coast but instead its spatial variability and the advection of air from warmer to colder waters which influences the occurrence of the stratocumulus cloud. Accordingly, the interplay of the dynamics of the Humboldt Current, atmospheric advection and coastal ocean upwelling as well as the large scale subsidence is essential for the existence of stratocumulus clouds at the coast of the Atacama.

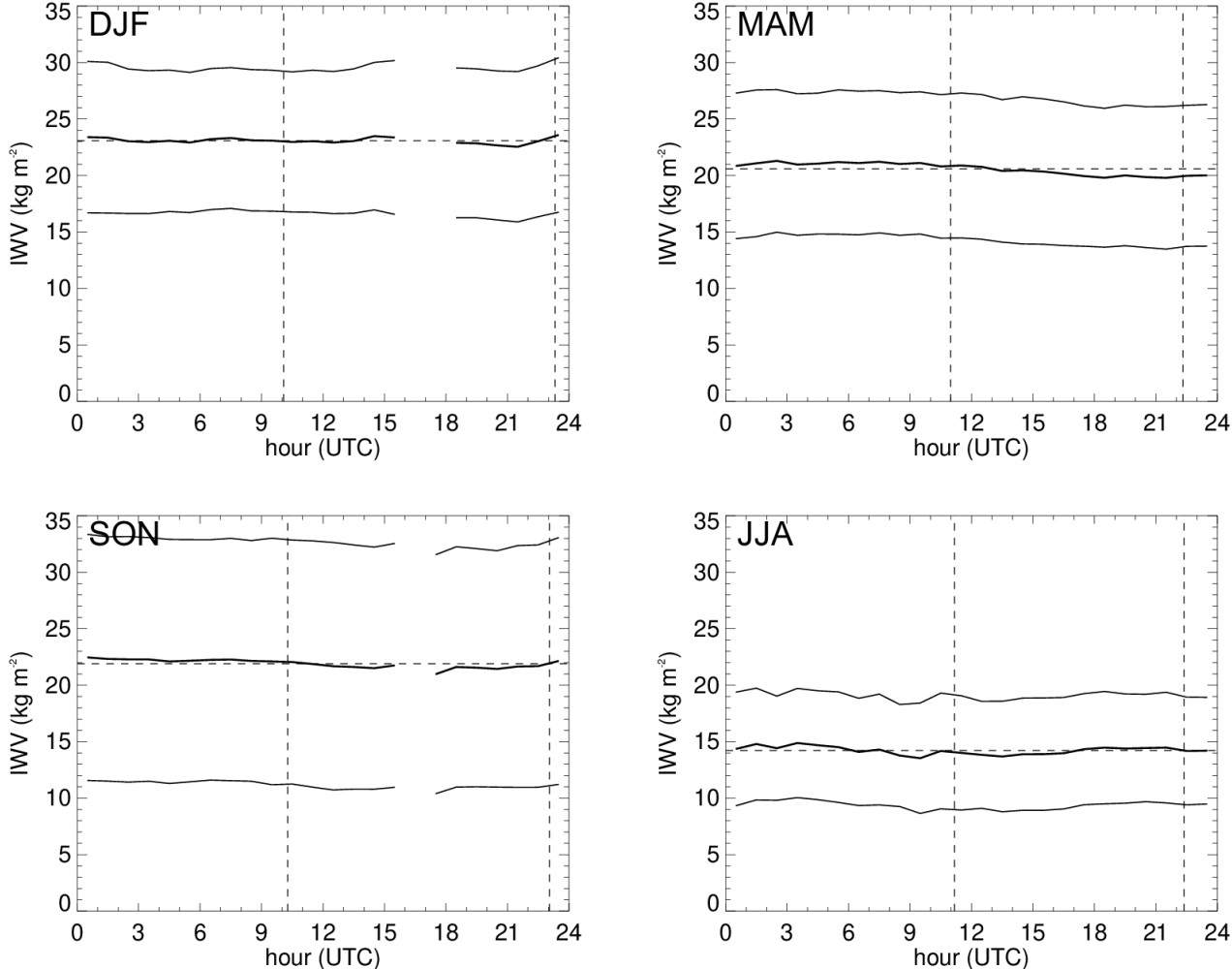

**Figure 16.** Average diurnal course of column integrated water vapor (IWV) during different seasons (thick line) plus-minus its standard deviation (thin lines) as a measure for day to day variability. Clockwise from top left: austral summer (DJF), fall (MAM), winter (JJA) and spring (SON). Vertical dashed lines indicate average times of sunrise and sunset. Gap around 17UTC in austral summer (DJF) is due to times when the sun is in zenith, making retrieval impossible.

## 4.2 Local Circulation and Diurnal Variation

We observed frequently that the stratocumulus cloud deck at the coast evaporate around noon and reappears in the afternoon or evening. The clouds evaporate from the bottom i.e. cloud-base height increases while cloud-top height remains constant (Fig. 7).

A rise of cloud-base may have two reasons: decrease of water content, or rise of temperature in the MBL. A decrease of water content could be explained via multiple steps: Absorption of solar radiation by the cloud during daytime reduces cooling at cloud-top and as a result the formation of plumes, responsible for the mixing in the MBL, would be weakened. This mechanism has been observed by Bretherton et al. (2004) above the open ocean. In the MWR data a weak reduction of the column Integrated water vapor (IWV) at the time of the onset of the sea breeze on the order of 0.5 kg/m$^2$ (Fig. 16) is observed

in all seasons. This is only a small part of the IWV but a large amount compared to the LWP values which lie in the range 0.01-0.1 kg/m$^2$ (Fig. 8). Especially in winter and spring the IWV recovers around 17-18UTC. This coincides with the time when clouds form again. Nevertheless, day to day variability of IWV at every hour is large and these diurnal variations might be just random. Additionally, it must be noted that IWV is the column integrated water vapor which means that entrainment of dry air from the free troposphere does not change the IWV.

As the MBL is, especially in winter, well mixed and neutrally stratified (Fig. 4) we can use 2 m temperature and humidity measurements to investigate the state of the MBL (Fig. 17), and lifting condensation level (LCL) to estimate cloud base. As a measure of the moisture content we use dew point. LCL is calculated by the linear estimate $LCL = 125 \, m/K \cdot (T_{\text{air}} - T_{\text{dew}})$ of Lawrence (2005) which is within $\pm 12$m with the exact formula of Romps (2017) for the observed temperature and humidity range. Air temperature shows a typical diurnal course with low temperatures during night, an increase from the early

morning hours and a maximum at 18UTC, i.e. in the early afternoon, which means that it is mainly defined by the radiative forcing by the sun. During night and early morning, moisture, represented by the dew point, shows a decrease towards a minimum at about 12UTC after which it starts to increase. This increase coincides with the onset of the sea breeze around 12UTC (see Fig. 3) and has its maximum rather late at 21UTC, 2 hours after the maximum of the sea breeze and 3 hours after the air temperature maximum. Thus, at the surface, we observe mainly a change between dry desert and moist ocean air which

is similar to observations in the coastal mountains (García et al., 2021) but also from other coastal stations (See data described in Schween et al., 2020).

The resulting LCL is constant during night, increases from sunrise until a maximum at around 16:30UTC, i.e local noon, and decreases again until sunset. It accordingly follows the course of air temperature in the morning hours, but decreases earlier in the afternoon. The reason for this afternoon behaviour is the increased moisture content which allow the reformation of clouds

in the afternoon.

Intercomparison between LCL and Cloudnet cloud base height (CBH, fig.17d) shows some cases with LCL < CBH, a strong cluster between 500 m and 700 m with LCL$\simeq$ CBH which coincide with daytime heights of LCL. But additionally there are several cases with LCL by up to 200 m hihger than CBH. Especially for CBH below 500 m appears a cluster with CBH by

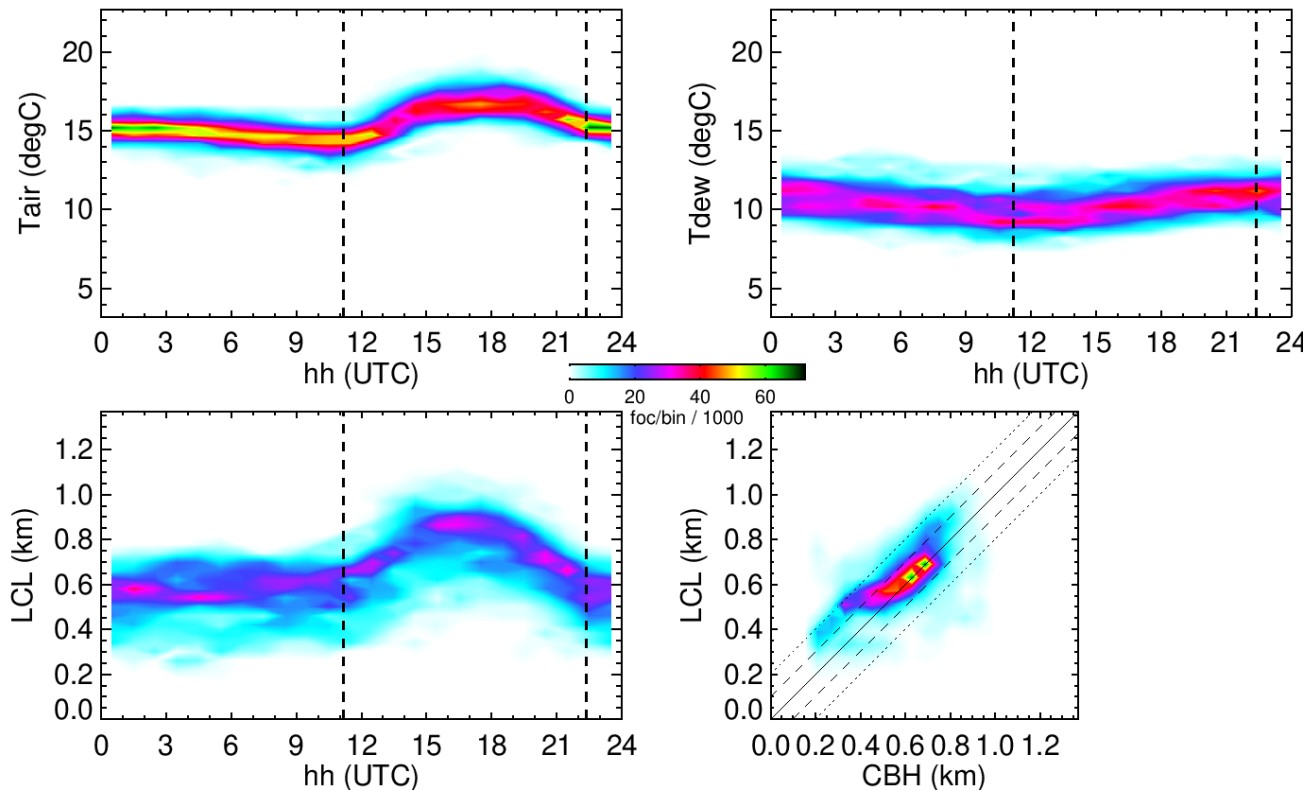

**Figure 17.** Air temperature at 2 m (top left, a), dew point (top right, b) and lifting condensation level (LCL, bottom left, c) versus hour, and LCL versus cloud base heights from Cloudnet (CBH, bottom right, d). All quantities during winter (JJA), presented as two dimensional frequency distributions. Vertical dashed lines indicate average times of sunrise and sunset respectively. Diagonal lines in the bottom right plot indicate ±100 m (dashed) and ±200 m (dotted) deviation from perfect agreement.

100 m-200 m lower than LCL. This is not possible in a mixed BL and even in a stratified BL it would require strong vertical humidity gradients.

The airport operates since 2017 a ceilometer (Vaisala CL31, see Direccion Meteorologica de Chile (DMC) Servicios Climaticos , 2022; del Rio et al., 2021b). A ceilometer provides 'ceiling', i.e. the largest height from which a pilot could see the earth surface, and especially Vaisala ceilometers do so by estimating visibility for a pilot using several assumptions (Martucci et al., 2010). In contrast hereto the CBH retrieval of Cloudnet (and some other ceilometer manufacturers) is based on the fact that cloud droplets show strong backscatter. CBH is then identified by a combination of the values of the backscatter coefficient $\beta$ and the occurrence of a strong vertical gradient in $\beta$ (For details of the Cloudnet CBH algorithm see Tuononen et al., 2019). As visibility decreases only above cloudbase, ceiling is typically larger than CBH with differences strongly depending on the cloud and cloud droplet properties (Martucci et al., 2010).

We compared CBH from the airport ceilometer with Cloudnet and saw a systematical underestimation by Cloudnet especially during winter and night with average differences around 100 m. Investigation of time series revealed that these large differences coincides with drizzle especially during night and morning hours. On the one hand this might be due to the fact that it is difficult to differentiate between drizzle and cloud droplets in the backscatter signal especially as drizzle originates from within the cloud (Tuononen et al., 2019). As a result CBH may indeed become too low. On the other hand drizzle increases humidity below the cloud and thus may lower cloud base. In this case LCL based on surface measurements is not valid for an estimation of cloudbase any more.

With this information we can describe the diurnal life cycle as follows: In the morning hours warming of the air from the surface evaporate the stratocumulus from the bottom. The upper branch of the sea breeze circulation transports the warm, cloud free air over the ocean and generates a cloud gap growing from the coast as can be observed nearly every day in satellite images. With its onset the surface branch of the sea breeze transports moist air towards the coast and therefore allows the reformation of clouds already in the afternoon although temperature is then around its maximum. The upper branch of the sea breeze pushes these clouds over the ocean. Eventually the coastal cloud connects to the maritime stratocumulus deck and after sunrise the process begins again.

We note here that the interaction of the land-sea-breeze with the 'Rutllant cell' as desribed in Rutllant et al. (2003) can not be inferred from the wind lidar measurements as the latter is restricted to the aerosol loaded MBL. The Rutllant cell comprises strong winds inland at altitudes around 1000 m above sea level and moisture transport into the hyperarid Atacama desert (Schween et al., 2020). If the cell would extend over the ocean strong wind shear would occur at the top of the MBL and could provide moisture from the MBL to the desert.

## 5 Conclusions

This paper presents ground-based remote sensing profiling observations of coastal stratocumulus clouds at the airport of Iquique at the northern coast of Chile. These clouds are a vital moisture supply for flora and fauna in the western part of the Atacama Desert.

The performed observations, for the first time, bring forth a full seasonal cycle of vertically resolved insights into the physical processes of the marine stratocumulus clouds interacting with topography of northwestern Chile. The observations resolve the cloud vertical structure, including cloud classification and microphysics, the turbulent structure of the ABL as well as the temperature. Additionally vertically integrated values of liquid water and water vapor have been recorded - all with a temporal resolution of seconds to minutes.

Clear seasonal and diurnal dependencies of cloud occurrence, geometrical extent as well as liquid water content have been observed (see Tab. 3). Compared to austral summer, stratocumulus clouds in austral winter occur 4.9 times more often, are 83% thicker (with 50 g/m$^2$ or 3.5 times more LWP), whereby cloud-base is 580 m lower. These differences are strongly related to the seasonal SST patterns off the Atacama Coast.

**Table 3.** Summary of the major findings of the present work during the Iquique airport observations (2018/2019) for austral winter (JJA) and summer (DJF). Values represent seasonal averages.

| parameter | Summer (DJF) | Winter (JJA) |
|---|---|---|
| surface pressure (hPA) | lowest (1007) | highest (1010.5) |
| SSTs @ coast (°C) | warmest (19.5) | coldest (16.5) |
| surface wind speed (m/s) | highest (7.5) | lowest (5.5) |
| cloud occurrence (%) | minimum (16) | maximum (79) |
| cloud-base height (m) | highest (1080) | lowest (500) |
| cloud thickness (m) | minimum (220) | maximum (400) |
| LWP ($g/m^2$) | minimum (20) | maximum (70) |
| IWV ($kg/m^2$) | highest (22.0) | lowest (14.1) |
| inversion height (m) | highest (1050) | lowest (950) |
| drizzle occurence (%) | lowest (1) | highest (18) |
| MBL stratification | stable | neutral |
| turbulence | mostly non-turbulent | more turbulent mixing |

The diurnal cycle shows a distinct pattern with minimum cloud occurrence (with lowest LWP) around 16 UTC, and maximum occurrence (with highest LWP) during night. Our observations show that at night clouds are maintained through turbulence originating from the cloud which connects them to the ocean surface. During daytime convective turbulence driven by the warm land surface frequently evaporates the stratocumulus cloud from the bottom. Clouds reestablish in the early afternoon as moister air is advected by the sea-breeze. The upper branch of the sea-breeze circulation drives these clouds over the ocean.

During summer and autumn, stratification in the MBL is typically stable while winter and spring shows neutral stratification and a well mixed MBL. This somewhat counter-intuitive behaviour results from the SST distribution in the ocean: while in winter water temperatures are over a wide range of distances constant, summer is characterized by coastal water temperatures 5 K lower than 50 km and more from the coast. Driver for this difference is a farther west-ward position of the Humboldt-current. This allows warmer tropical waters to reach further south during these seasons while coastal upwelling maintains a low SST along the coast. The MBL appears to be in equilibrium with these warmer waters and becomes stably stratified when advected towards the coast which, in turn, makes clouds less probably to persist.

Especially in spring, cloud cover at the north coast of Chile seems weakly connected to the El-Niño ONI 3.4 index, i.e. SST anomaly in the Equatorial Pacific several thousand km to the north-west (del Rio et al., 2021a). Our observations may help to understand the details of this coupling and allow predictions of future and past occurrence of stratocumulus clouds at the coast. Based on our observations it will be possible to investigate details of the processes by use of large eddy simulation (LES).

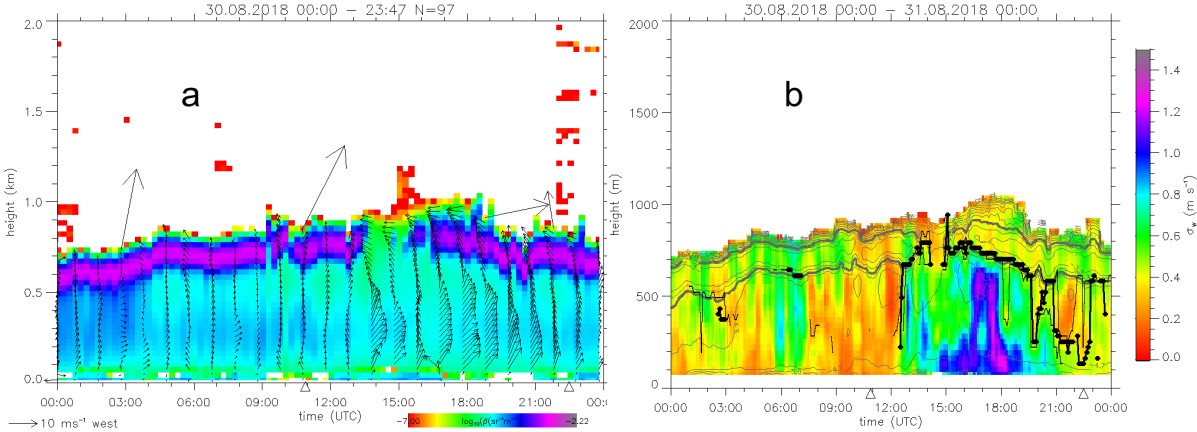

**Figure A1.** Example data from August 30, 2018. Left (a): Horizontal wind speed and direction (vectors) and attenuated backscatter ($\beta$, color shading) as a function of time and height Right (b): Standard deviation of vertical velocity from the wind lidar (color shading), backscatter ($\log_{10}\beta$, thin gray lines), with $\log_{10}\beta = -5$ as a typical value for cloud base (thick gray line) and mixing layer height estimate (after Schween et al., 2014, thick black line with symbols). Note that the upper border of the region with backscatter is due to extinction of the laser signal and does not represent cloud top. Triangles at the abscissa indicate times of sunrise and sunset with night to the left and day on the left side.

*Data availability.* Data from wind lidar, microwave radiometer, cloud radar and meteorological station as well as the synergisitc boundary layer classification is accessible via the SFB 1211 'Earth - Evolution at the Dry Limit' projects data base webpage https://www.crc1211db.uni-koeln.de. Basic meteorological data is available under DOI 10.5880/CRC1211DB.45. Wind profiles are available under DOI 10.5880/CRC1211DB.53. Temperature profiles are available under DOI 10.5880/CRC1211DB.46. LWP for the different relative humidity thresholds are available under DOIs 10.5880/CRC1211DB.49 (TH80), 10.5880/CRC1211DB.50 (TH85), 10.5880/CRC1211DB.51 (TH90), 10.5880/CRC1211DB.52 (TH95). IWV is avaiable under DOI 10.5880/CRC1211DB.43. The boundary layer turbulence classification is available under 10.5880/CRC1211DB.54.

The synergistic Cloudnet classification data used in this article was generated by the European Research Infrastructure for the observation of Aerosol, Clouds and Trace Gases (ACTRIS) and are available from the ACTRIS Data Centre using the following link: https://hdl.handle.net/21.12132/2.e224164deb7c40c5.

## Appendix A: Example Case 30.8.2018

Figures A1 - A3 show example data from all three instruments and the cloudnet retrieval for August 30, 2018, a typical winter day with continuous cloud cover during most of the time, a cloud gap around noon and drizzle during the night.

Wind profiles (Fig.A1) show during daytime strong south-westerly winds at heights below about 500 m, and above a continuous turn to south-easterly and easterly winds. During night wind changes between south-west and south-east which may represent circulations induced by the falling drizzle. High attenuated backscatter $\beta$ between 500 m and 800 m identifies the lower part of the cloud. Inside the cloud the signal decrease due to extinction and a wind retrieval is possible only up to 200 m

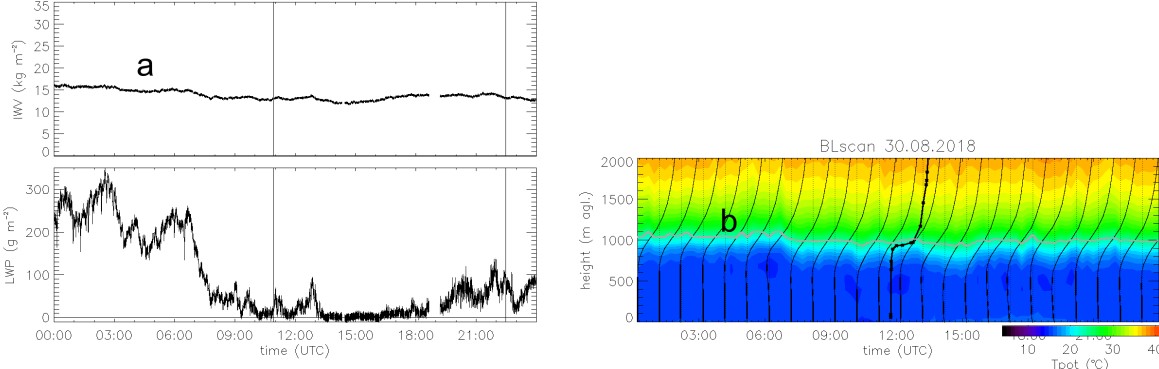

**Figure A2.** Example data from August 30, 2018. Left (a): LWP and IWV from the microwave radiometer on basis of the TH85 retrieval. Right (b): Potential temperature profiles from the microwave radiometer (color shading, and thin smooth lines at times indicated by dotted vertical lines) and the Antofagasta radiosonde profile for that day (thick line with symbols around 12UTC). The thick gray line indicates the height of the strongest gradient. Vertical lines indicate times of sunrise and sunset.

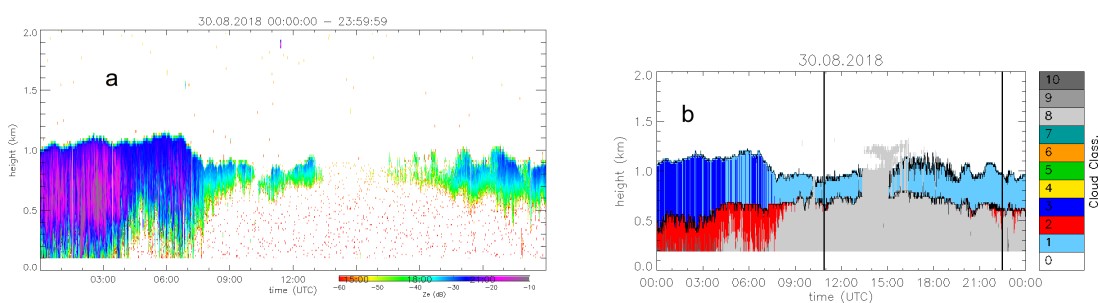

**Figure A3.** Example data from August 30, 2018. Left (a): Radar reflectivity Ze from cloudradar. Right (b): Cloudnet classification with classes clear sky (0, white), cloud droplets only (1, light blue), drizzle or rain (2, red), drizzle or rain and cloud droplets (3, dark blue) and aerosol (8, light gray), thin black line indicates cloud base and cloud top. Vertical lines indicate times of sunrise and sunset.

into the cloud but single implausibly large vectors indicate problems of the retrieval in this region. The wind presented here
is based on a scan which takes about 1 minute , i.e. it represents the instantaneous wind, and fluctuations in time represent its variability. Wind lidar backscatter data (A1) show that the cloud vanishes at 1330UTC and reappears at 15UTC. Between 1230UTC and 1930UTC more intense turbulence with vertical velocity standard deviation $\sigma_w > 0.4$ reaches up to cloud base, which during this time is higher than during hours before and after.

Integrated water vapor (Fig.A2) shows weak variation over the day with highest values around 15 kg/m$^2$ in the first half
of the night before 6UTC, and lowest values of 12 kg/m$^2$ around 15UTC when the cloud disappears. Liquid water content (LWC) is high before 7UTC, decreases to lower values in the hours before 13UTC and remains for the time until 19UTC to values close to zero. After 19UTC values reach the same level as in the morning hours. Temperature profiles show between 1230UTC and 20UTC super adiabatic stratification in the lowest hundred meters coinciding with the turbulence visible in the

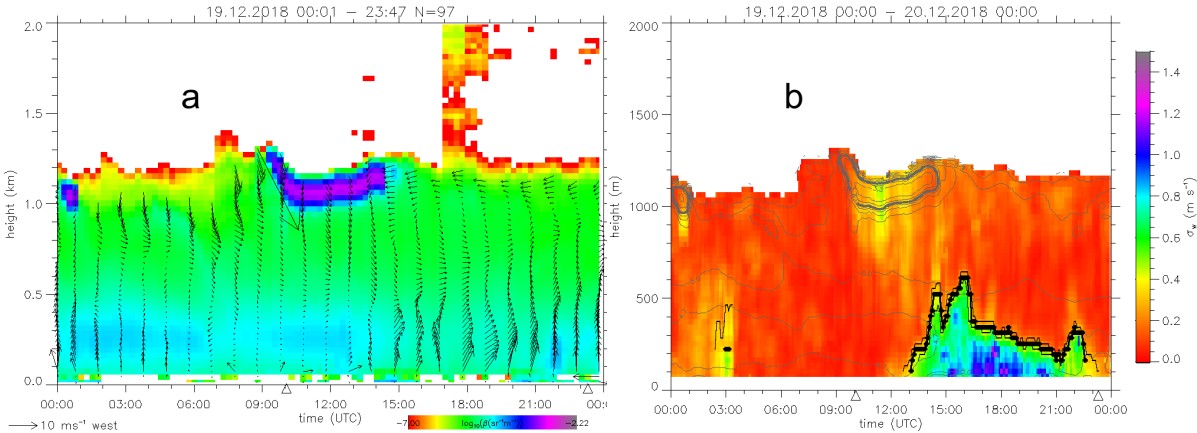

**Figure B1.** Like Fig. A1 for December 19, 2018.

vertical velocity data. Although the gradient of the inversion in the MWR profiles is much weaker and the inversion layer is much broader than in the radiosonde data, the height of the 'MWR-inversion' is about the same height as the inversion visible in the Radiosonde data. Inversion height shows only a weak variation over the day which indicates that it is neither influenced by subsidence nor divergence in the land-sea-wind circulation.

Cloud radar reflectivity (Ze, Fig.A3) show values during the night with structures indicating drizzle. At 8UTC drizzle stops and the remaining cloud shows much lower Ze values. At 13UTC the cloud disappears in the radar echo and reappears at 19UTC. As we can infer from the wind lidar backscatter the cloud was nevertheless present in the afternoon and the radar is not sensitive enough to detect the cloud during this time.

Cloudnet combines the data from the different instruments and operational model data, identifies cloud boundaries and classifies the cloud (Fig.A3). Presence of a cloud is inferred from significant lidar and radar backscatter as well as nonzero liquid water content from the microwave radiometer. Cloudbase is identified from lidar backscatter on the basis of backscatter and its gradient. Cloudtop is based on the top of the region with nonzero radar reflectivity. If the lidar provides a cloud base but the radar detected no cloud, cloud base is inferred from the lidar and cloud top is calculated from the theoretical adiabatic liquid water content gradient and LWP from the MWR. Cloudnet finds drizzle before 8UTC, a strong drop of cloud top with end of the drizzle, a cloud gap between 13UTC and 15UTC, and when present a cloud base height with a maximum at 16UTC and a clear decrease afterwards which coincides with a decrease of lifting condensation level.

## Appendix B: Example Case 19.12.2018

Figures B1 - B3 show example data for December 19, 2018, a typical summer day with one cloud present in the early morning, stable stratification during night and convective turbulence reaching up to less than half of the MBL height.

Wind profiles show again during daytime stronger south-westerly winds in the lower part of the MBL and easterly winds in the upper part (Fig. B1). Night time winds in the lower half of the MBL are weak with southerly to easterly directions,

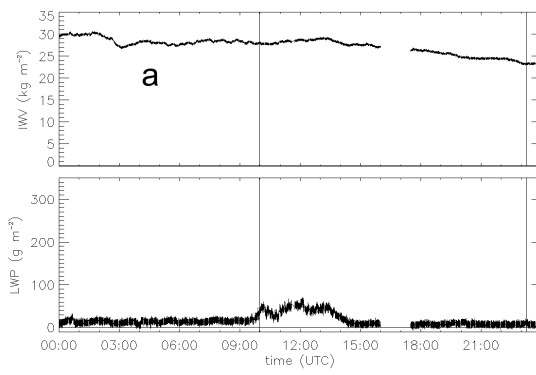

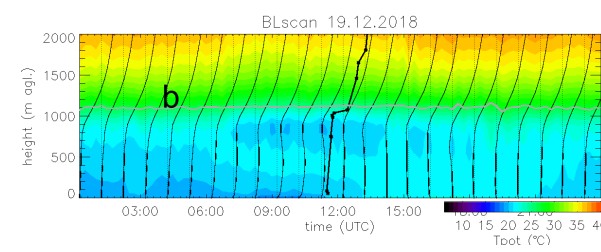

**Figure B2.** Like Fig. A2 for December 19, 2018.

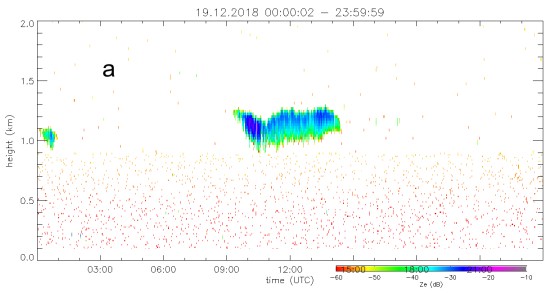

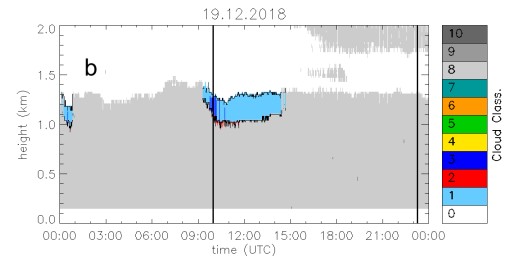

**Figure B3.** Like Fig.A3 for December 19, 2018.

while above stronger westerly and northerly winds prevail. The mixed layer with vertical velocity standard deviation larger than 0.4 m/s reaches less than half of the MBL height. High backscatter above 900 m between 9UTC and 15UTC indicates a cloud which is the source for slightly increased turbulence below reaching downto 700 m.

    MWR data reveal a low LWP around 40 g/m$^2$ for this cloud (Fig.B2). Temperature profiles show during night and morning stable stratification which can also be found in the radio sounding from Antofatgasta for this day. This stable stratification

disappears around 13UTC and even super adiabatic stratification close to the surface becomes visible. The height of the MWR inversion remains over the whole day at the same height indicating that it is neither influenced by ground based convection, nor subsidence or divergence due to the land-sea-breeze circulation. Cloud radar reflectivity (Fig. B3) shows the cloud and cloud net retrieval identifies it as a pure liquid water cloud. Its top coincides with the top of the aerosol layer (1300 m) which accordingly lies above the MWR-inversion (1100 m). Cloud net identifies in the 2 hours after sunrise before 12UTC drizzle for

some few pixels which is also the time with the strongest turbulence below the cloud. Nevertheless enhanced turbulence below the cloud is also present in the following hour indicating cooling by evaporation at cloud top generating subsiding plumes.

*Author contributions.* UL, CdR, JLG and PO planned the campaign and selected the site. SW and JHS did the data analysis. UL, SW and JHS planned and structured the paper. JHS wrote the manusript. JHS, CdR, JLG, PO, SW and UL reviewed it iteratively.

*Competing interests.* The authors declare that they have no conflict of interest.

*Acknowledgements.* The project has been funded by the Deutsche Forschungsgemeinschaft (DFG, German Research Foundation) Collaborative Research Centre 1211 Earth - Evolution at the Dry Limit (SFB 1211, Projektnummer 268236062). Data analysis procedures and retrievals are developed at the University of Cologne through the joint efforts at CPEX-LAB (www.cpex-lab.de). We are grateful to Anne Hirsikko and Prof. Heikki Lihavainen from the Finnish Meteorological Institute (FMI) for providing the Doppler lidar for this installation. We are grateful to the Dirección General De Aeronáutica Civil, Departamento Comercial (DGAC) for giving their allowance to install the
instruments on the grounds of Diego Aracena airport at Iquique, and Marcial Vidal Arriagada for support from side of the Airport management. We thank Clara Stock, from the Centre for Organismal Studies, University Heidelberg, and Constanza Vargas from the Centro UC Desierto de Atacama, for maintenance and technical support of the instruments during their installation at Iquique airport. We acknowledge the ACTRIS Cloud Profiling Unit (CLU) for providing the dataset in this study, which was produced by the Finnish Meteorological Institute, and is available for download from https://cloudnet.fmi.fi/. We especially thank Ewan O'Connor from FMI to include the instruments in
the Cloudnet operational processing and Tobias Marke for setting up the turbulence classification scheme. This work would not have been possible without our technicians Pavel Krobot and Rainer Haseneder-Lind and their efforts in preparation, packing and sending the instruments, setup and maintenance of the internet connection and support for the way back. We also thank Susanne Crewell and Thomas Rose for assisting during the instrument setup. Finally, the authors are very grateful to the efforts of Mario Mech, whose experienced support in preparing and carrying out the setup of he measurement station was indispensable for the success of the project.

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
