# Peer review of "Life Cycle of Stratocumulus Clouds over one Year at the Coast of the Atacama Desert"

_Atmospheric Chemistry and Physics, 2022_

## Referee Comment (RC1)

Review of ACP-2022-108

Overview:

The authors document 1 year measurements of the coastal ABL at Iquique, Chile, on the eastern border of the extensive SW Pacific stratocumulus region. The measurements include a cloud radar, a microwave radiometer and a Doppler wind lidar. This suite of instruments permits a very complete characterization of the thermodynamic and dynamic evolution of the regional lower troposphere, including the seasonal and diurnal variation, which are the focus of the paper. The text is generally well written and the results represent a valuable contribution to the field. I recommend acceptance after the major and minor comments below are properly addressed.

Recommendation: major revision

Major comments

1. Water loss at the top of the MBL

In several parts of the paper the authors indicate that water of the MBL is lost by evaporation into the free troposphere. I take issue with this concept which I believe is wrong. At the top of the MBL the main process is entrainment. In this process, free-tropospheric air is entrained into the MBL, and the MBL grows through this mechanism. It is true that the MBL dries because of entrainment but this does not represent a net loss of water for the MBL. To support my comment more quantitatively, let me consider 2 of the basic equations that represent a mixed layer evolution:

$$\frac{dq}{dt} = \frac{\overline{w'q'}_o - \overline{w'q'}_H}{H} \qquad (1)$$

$$\overline{w'q'}_H = -(q_+ - q)\frac{dH}{dt} \qquad (2)$$

where q is the MBL water mixing ratio, H is its depth, $q_+$ is water mixing ratio in the free troposphere and the covariances denote the turbulent water fluxes at the surface (subscript o) and at the MBL top (subscript H). These 2 equations are considered exact in the frame of mixed layer modeling (i.e. they do not include any parameterization assumption). In a typical situation, $q_+ < q$ and the water turbulent flux at the top is positive (eq. 2), which induces a decrease in q (eq. 1). This positive flux at the MBL top is what the authors appear to interpret as a water loss for the MBL. However, the total water content in the MBL is qH, for which we can derive an evolution equation by combining both equations above:

$$\frac{d(qH)}{dt} = \overline{w'q'}_o + \overline{w'q'}_H \left(\frac{q_+}{q - q_+}\right)$$

From this equation it is clear that the entrainment is a source of water to the MBL and not a loss. In 1-D equilibrium, the water loss process should be a precipitation mechanism (drizzle) and eventually subsidence and horizontal divergence.

2. Comparison of measurements with standard measurements at the airport and with M2016

The authors provide a wealth of new data about the MBL at Iquique but make little effort to put their data in connection with the existing operational data at the airport: surface meteorology and cloud base and cloud fraction. They therefore miss an opportunity of validating the airport and their data as well. This is especially needed when they find a substantial difference between LCL and cloud base, which calls for a more critical analysis of the data. The same can be said about the comparisons with the results of Muñoz et al. (2016). I believe the authors can be more explicit and critical in performing this comparison.

3. Description of seasonal and diurnal cycles.

The paper focuses on the diurnal and seasonal cycles of the data. However, I find that their figures representing this variability can perhaps be enhanced. See comments later.

4. Checking averaged evolution against specific cases

The authors present mostly averaged and climatological results with just 1 figure showing a particular case (Figure 2) with too little discussion of their measurements on that day. All their discussion is based on the average conditions. I'd like to see more evidence that individual cases are faithfully represented by the averages. For example, the case of Figure 2 does not show very clear the dissipation by cloud base growth that is present in the average fields.

Detailed comments

L15: Please add a reference to support your first sentence.

L16: Add comma after radiation

L18: equatorward

L21: What do you mean by "stabilized"? some of these mechanisms could produce the dissipation of the clouds (e.g. CTEI).

L24: Parenthesis missing in the references.

L25: The subject of the sentence is "stratocumulus" which is a singular noun, hence the correct verb is "has"

L28: ... and, accordingly, it determines...

L44: Replace "turbulence" by "eddies".

L46: ... entrainment, a too low ...

L64: Eliminate parenthesis around references. Replace ; by and

L73: ... waters, cloud cover...

L79: Latitude and longitude are incorrect

L79: What is DFG?

L101: What do you mean by "larger parts of the coast"? Please rephrase.

L114: Remove extra parenthesis

L134: The authors are too vague in describing the expected performance of the instrument: "to some extent" and "coarse vertical resolution". Please be more specific and quantitative.

L142: How site-specific is the calibration methodology of the MWR? The fact that you use radiosondes far from Iquique to perform the calibration introduces uncertainty in your results? Please comment.

L143-154: I believe this paragraph is better put at the end of section 2.2.2

L163: Please be explicit and quantitative on the vertical resolution attained by the instrument.

L190: It would be nice if you can show the position of these 2 points in Figure 1.

L205: I was confused by the terminology "Boundary layer classification". I was expecting different classes of BL, but even in 1 BL case the scheme classifies different layers of the BL in different categories. Hence I believe a better name would be "turbulent layers classification" or "turbulence classification".

L206: Cloudnet

L208: Only below the cloud? I get the impression you included the cloud layer as well.

L239: Figure 2 includes a panel with the cloudnet classification for the day. However you do not discuss it at all. In particular, it is curious that this specific case does not show the "averaged" behavior of the dissipation. The cloud base does not rise much during the morning and at some time the cloud layer suddenly disappears (see major comment). I'd suggest that for this example day, you add a new figure with measurements of your 3 instruments and the cloudnet classification and expand the analysis and discussion. In this manner, the reader can assess how well the averaged fields presented later represent individual cases.

Figure 2: In the cloudnet panel please mark the times of the satellite pictures. The legend text is not readable.

L240-L260: As the authors indicate, winds in Figure 3 show a marked diurnal cycle and subtle seasonal variation. Therefore, I believe that the representation in Figure 3 could be enhanced. For example, show just 1 diurnal cycle (e.g. SON) and then show the difference of the diurnal cycles between 2 periods (JJA-DJF). This will make clearer the seasonal changes. Alternatively, show the annual diurnal cycle and the monthly variation of the 19-21 UTC and the 06-08 UTC winds corresponding to the extremes of the diurnal cycle.

Figure 3: Please explain somewhere the reason of the reduced number of cases of some of the periods. Indicate if height is ASL or AGL. Also, please indicate whether all data in the figures is the average of the same number of days or the borders have less data. In the latter case, the robustness of the averages in the border should be discussed.

L262: Please indicate how the potential temperature was computed.

Figure 4: Please show the temperature fields instead of the potential temperature fields. The most distinct feature of the thermal structure in the region is the prominent subsidence inversion which is most clear from temperature fields.

Figure 4: The same suggestions for Figure 3 apply here. Please explore ways of showing the seasonal and diurnal variation of the fields more distinctly.

Figure 4: Please indicate number of days used in the averages in this case.

L271: The height of the maximum temperature gradient should be compared with the average of inversion base and inversion top. In this case the comparison with the results of Muñoz et al. 2011 is not so good, especially considering that IQQ is to the north of ANF. Please discuss.

L276: The near surface stability is probably strongly conditioned by the land surface where the measurements are made and therefore they do not necessarily describe the conditions above the sea and the argument about the surface fluxes is weak. Please discuss.

Figure 5: Please express the FOC as percentages in the figure so that the discussion in L280-281 is better followed.

L287-288: Please check whether this feature of the morning dissipation is due to the averaging or indeed happens in most of the days (see comment of Figure 2).

L298: Actually, I think the comparison with Muñoz et al. (2016) is not as good. The authors should make the comparison in more detail and compare also their results with the operational measurements performed at Iquique airport (see major comment).

L305: "water loss by evaporation at cloud-top". The authors must be careful about what they mean with this phrase. Do they refer to liquid water or to total water? In my view, entrainment at the top of the cloud layer does not imply total water loss for the MBL, as the entrained air is intruded into the MBL. Liquid water can be lost due to evaporation increased by the dry air entrained. It is not clear what exactly the authors mean here. Please explain and see also major comment.

L310: It appears that figure 8 is not discussed in the text.

L314-315: The phrase about the uncertainty of the LWP retrieval is not clear. Please provide an explanation of the uncertainty mentioned (20-30 ug/m2). And if this is the case, then figure 9 should in some way convey that huge uncertainty in the plotted values. As it is now, it is

misleading because the uncertainty appears to be described by the 4 retrievals which are almost identical.

L316: It is noteworthy that in JJA the maximum LWP is found very early in the night and values decrease thereafter. Please discuss that.

L319: Eliminate "The height of".

Figure 9. I'd have expected that TH80 retrieval always produced larger LWP than TH95, but this is not the case in the afternoon, especially winter and spring. Can you comment on that?

Figure 9: Please indicate variability and/or uncertainty of these figures.

L331: frequency

L338: I suggest that the authors make an effort to analyze in more detail the drizzle occurrence, considering that they have so much data available. Scatterplots of day-to-day indices of drizzle, cloud depth and LWP are missed. For example, the simplest parametrizations of drizzle in Sc clouds is to define a threshold in maximum LWC above which drizzle begins. As the authors have estimates of LWP and cloud base and top heights, such relationships can be tested. I strongly suggest that they explore such relationships.

L343: See my concern with the "BL classification" terminology.

L361: What do you mean by "counter-intuitive"?  I suggest the adjective "contrasting" or be more specific about the intuition behind the comment.

Figure 12: Can you separate your data in Cloudy/Clear nights and see how these diurnal classification changes? I believe that such analysis can provide more physical insights.

L366-367: Again the argument of water loss by evaporation. Liquid water perhaps, but entrainment of tropospheric air into the MBL does not imply water loss for the MBL but the opposite. Please see major comment.

L376: in the cloud layer mixing produces moist-neutral conditions which are different than the dry-neutral conditions in the mixed CBL.

L385: I miss a consideration of subsidence, especially if the authors are trying to describe some type of equilibrium condition. I believe no equilibrium will be attained if subsidence is not included in the analysis, as turbulent entrainment always tends to deepen the MBL. Also, in this section drizzle is not mentioned although it must be important in an equilibrium analysis.

L407: There is ample literature on the relationship between lower-tropospheric stability and Sc-capped MBL, which should be cited.

L410: Again the idea of water loss at the top of the MBL. See major comment.

L412: and again.

L417: and again.

L422: All the analysis in this section endeavors to show a cause-effect relationship between SST, subsidence and clouds. However, as the authors are discussing long-term averages I see a conceptual difficulty in establishing a cause-effect relationship. In my opinion, the most can be said is that there is an association between the SST, subsidence and cloudiness in these averages.

L425: Please see a previous comment on the "dissolution from the bottom" effect in the averages in relationship to the fourth panel of figure 2.

L427: Eliminate extra "in the"

L434: What do you mean by "on the order of the values" ? what values?

L436: Again the concept of water loss to the free troposphere.

L437: Do you mean "winter"?

L440: Please correct the "dew" subindex.

L441: The 400 m difference is huge and deserves more analysis. The 3.2 K surface super adiabatic condition needed to explain the difference conflicts with the cloudy conditions and the overall assumption of the paper that their measurements represent the coastal conditions over the sea surface. The authors should make an effort to compare their results with measurements at the airport (cloud base and surface variables), as well as LCL and cloud bases reported previously by Muñoz et al. (2016).

L456: stratocumulus

L456: An alternative hypothesis for the daytime coastal clearing can be induced subsidence by the strengthening of the westerly component of the coastal wind during the day. I believe the data analysis of the authors is not sufficient to discard this hypothesis which has no need of an upper branch in the circulation. Their measurements of this circulation are only marginal as shown in figure 3.

L462: The term "Rutllant cell" was coined in a paper by Houston.

L462: "cannot be inferred"

L463: Replace "as it is " by "as the latter is"

L466: A scatter plot between LCL and cloud base would be of interest to appreciate their relationship beyond mean conditions.

Figure 15: The quality of this figure is substandard. Some panels show the variability and others do not or do so partially. Ranges of the vertical axes are all different, which makes comparisons difficult.

Figure 16 caption: two dimensional frequency distributions

---

## Author Comment (AC1)

**1 Reply to Reviewer 1**

**1.1 Overview:**

The authors document 1 year measurements of the coastal ABL at Iquique, Chile, on the eastern border of the extensive SW Pacific stratocumulus region. The measurements include a cloud radar, a microwave radiometer and a Doppler wind lidar. This suite of instruments permits a very complete characterization of the thermodynamic and dynamic evolution of the regional lower troposphere, including the seasonal and diurnal variation, which are the focus of the paper. The text is generally well written and the results represent a valuable contribution to the field. I recommend acceptance after the major and minor comments below are properly addressed. Recommendation: major revision

*We thank reviewer 1 for carefully reading the text and giving his valuable arguments. We will adress them point by point with reply in italic text, and* new or revised text in red .

**1.2 Major comments**

**1.2.1 Water loss at the top of the MBL**

In several parts of the paper the authors indicate that water of the MBL is lost by evaporation into the free troposphere. I take issue with this concept which I believe is wrong. At the top of the MBL the main process is entrainment. In this process, free-tropospheric air is entrained into the MBL, and the MBL grows through this mechanism. It is true that the MBL dries because of entrainment but this does not represent a net loss of water for the MBL. To support my comment more quantitatively, let me consider 2 of the basic equations that represent a mixed layer evolution:

$$\frac{dq}{dt} = \frac{\overline{w'q'}_o - \overline{w'q'}_H}{H} \tag{1}$$

$$\overline{w'q'}_H = -(q_+ - q) \cdot \frac{dH}{dt} \tag{2}$$

where $q$ is the MBL water mixing ratio, $H$ is its depth, $q_+$ is water mixing ratio in the free troposphere and the covariances denote the turbulent water

fluxes at the surface (subscript $o$) and at the MBL top (subscript $H$). These 2 equations are considered exact in the frame of mixed layer modeling (i.e. they do not include any parameterization assumption). In a typical situation, $q_+ < q$ and the water turbulent flux at the top is positive (eq. 2), which induces a decrease in q (eq. 1). This positive flux at the MBL top is what the authors appear to interpret as a water loss for the MBL. However, the total water content in the MBL is $q \cdot H$, for which we can derive an evolution equation by combining both equations above:

$$\frac{d(qH)}{dt} = \overline{w'q'}_o + \frac{q_+}{q - q_+} \cdot \overline{w'q'}_H \tag{3}$$

From this equation it is clear that the entrainment is a source of water to the MBL and not a loss. In 1-D equilibrium, the water loss process should be a precipitation mechanism (drizzle) and eventually subsidence and horizontal divergence.

*This is of course an idealized model and indeed exact under certain circumstances like a well mixed boundary layer which can be characterized by one single total water mixing ratio q including water vapor and liquid water, an infinitely sharp separation between boundary layer air and free tropospheric air characterized by $\Delta q = q - q_+$, horizontal homogeneity and accordingly no horizontal advection, no subsidence, etc.*

*Nevertheless with equation 2 the reviewer assumes that the height of the moist layer coincides with height of the BL, and any transport of moisture across BL-top expands the height of the BL (see Fig.1). This model intrinsically assumes that no moist air is lost to the free troposphere, i.e. $\overline{w'q'}_+ = 0$. And of course this assumption results in an increase of column water content qH of the BL when moisture is transported into the region just above the BL. With the puzzling result that for $q_+ = 0$ the entrainment flux does not contribute to the column water content. The reason is that in this case no water from the free troposphere can contribute to the water content of the BL.*

*In these kind of budget equations for the BL it is usually assumed that the entrainment flux is related by some fraction k to the surface flux (see e.g. Lilly, 1968 or Schubert et al 1979 or Oversloot 2013 ). A more sophisticated model would have to consider that a detrainment into the free troposphere would increase moisture content in the layer just above H without making it directly a part of the BL. This increase would depend on diffusivity in the free troposhere which would require additionally parameters like wind shear and*

[Figure]

Figure 1: Schematic on equation 2. Black line is the $q$-profile in a BL with height $H$. Red lines indicate a moistening of a layer of thickness $dH$ on top of the BL after a time step $dt$. Black arrows indicate the entrainment of moist air into the dry free troposhere which becomes thus part of the BL.

*temperature gradient in the inversion and above which is beyond the scope of this type of models.*

*Equation 2 might be a good approximation for a convectively growing boundary layer where warm and moist plumes reach up into the free and dry troposphere and thus mediate at the same time a growth of the mixed layer and a moistening of the layer. This equation neglects that with a constant BL height $H$ there might be entrainment of dry air into the BL which must be compensated by detrainment of moist air into the free troposhere. Both processes together give in total a loss of moisture to the free troposhere i.e. a positive $\overline{w'q'}_H$. But this does not necessarily mean that the BL grows. In our data we observe no significant change of BL-height with time of the day (see fig. 4 in the original text). Consequently we believe that there is exchange between the free troposphere and the BL.*

*CHECK: can we see $\sigma_w \neq 0$ at BL-top in the Wili and/or radar data ? can we include this in the paper ?*

*For our discussion about the presence of clouds and their relation to the moisture content of the BL it is not the water column content $qH$ of the BL but rather $q$ which defines whether clouds can form and were their base is.*

*It is true that drizzle, horizontal divergence and subsidence may contribute to the development of the BL and thus also the water content of the BL. In the case of drizzle it changes q only if it reaches the ground as in this case water is remove from the BL. In our data we see frequent drizzle which does not reach the ground as has been already discussed in section 3.6.*

**Where necessary we changed the text to point to possible mechanisms for drying or moistening the BL.**

**1.2.2 Comparison of measurements with standard measurements at the airport and with M2016**

The authors provide a wealth of new data about the MBL at Iquique but make little effort to put their data in connection with the existing operational data at the airport: surface meteorology and cloud base and cloud fraction. They therefore miss an opportunity of validating the airport and their data as well. This is especially needed when they find a substantial difference between LCL and cloud base, which calls for a more critical analysis of the data. The same can be said about the comparisons with the results of Munoz et al. (2016). I believe the authors can be more explicit and critical in performing this comparison.

*We compared our data to the airport data. Whereas we found good agreement within instruments accuracy for air temperature and relative humidity. We realize that ceiling from the airport ceilometer and cloud base from the cloudnet algorithm show some systematic differences.*

*The comparison between the airport meteorology and our measurements is discussed in the description of the meteorological measurement (Sect. 2.2.4) as follows :*

Intercomparison of air temperatures ($T_a$) from our instruments and the operational airport measurements (DMC, 2022) about 1.5 km to the south, reveals good agreement with differences within $\pm 0.5$ K. An exception from this is summer when temperatures rise above about 22⁰C and our $T_a$ is on average 0.5⁰higher. This excess can be explained by the lack of an active ventilation in our measurements. Dew point calculated from our instruments is systematically 0.8 K lower than from the airport data with scatter in the order of 0.3 K. This is equivalent to by 3% too low readings in our relative humidity measurements. Although his is within the range of the instruments accuracy, we corrected this difference.

*The discrepancies between LCL and CBH and their relation to the airport data are discussed in detail in Section 4.2.*

Intercomparison between LCL and Cloudnet cloud base height (CBH, fig.16d) shows some cases with LCL ¡ CBH, a strong cluster between 500 m and 700 m with LCL≃ CBH which coincide with daytime heights of LCL. But additionally there are several cases with LCL by up to 200 m hihger than CBH. Especially for CBH below 500 m appears a cluster with CBH by 100 m-200 m lower than LCL. This is not possible in a mixed BL and even in a stratified boundary it would require strong vertical humidity gradients.

The airport operates since 2017 a ceilometer (Vaisala CL31, see DMC 2022, del Rio et al. 2021a). A ceilometer provides 'ceiling', i.e. the largest height from which a pilot could see the earth surface, and especially Vaisala ceilometers do so by estimating visibility for a pilot using several assumptions (Martucci et al. 2010). In contrast hereto the CBH retrieval of Cloudnet (and some other ceilometers) is based on the fact that cloud droplets show strong backscatter. CBH is then identified by a combination of the values of the backscatter coefficient $\beta$ and the occurrence of a strong vertical gradient in $\beta$ (For details of the Cloudnet CBH algorithm see Tuononen et al. 2019). As visibility decreases only above cloudbase, ceiling is typically larger than CBH with differences strongly depending on the cloud and cloud droplet properties (Martucci et al. 2010).

We compared CBH from the airport ceilometer with Cloudnet and saw a systematical underestimation by Cloudnet especially during winter and night with average differences around 100 m. Investigation of time series revealed that these large differences coincides with drizzle especially during night and morning hours. On the one hand this might be due to the fact that it is difficult to differentiate between drizzle and cloud droplets in the backscatter signal especially as drizzle originates from within the cloud (Tuononen et al. 2019). As a result CBH may indeed become too low. On the other hand drizzle increases humidity below the cloud and thus may lower cloud base. In this case LCL based on surface measurements is not valid for an estimation of cloudbase any more.

**1.2.3   Description of seasonal and diurnal cycles.**

The paper focuses on the diurnal and seasonal cycles of the data. However, I find that their figures representing this variability can perhaps be enhanced.

See comments later.

*We found the suggestion by the reviewer interesting but had to realize that especially due to the varying height of the boundary layer a normalization of the height coordinate would have been necessary making additional plots and explanations necessary. For a concise description we therefore stay wit the plots as they are (see also replies below).*

**1.2.4 Checking averaged evolution against specific cases**

The authors present mostly averaged and climatological results with just 1 figure showing a particular case (Figure 2) with too little discussion of their measurements on that day. All their discussion is based on the average conditions. I would like to see more evidence that individual cases are faithfully represented by the averages. For example, the case of Figure 2 does not show very clear the dissipation by cloud base growth that is present in the average fields.

*We found selected several plots and put them together in one figure. Regarding the current size of the document we decided to put them in the appendix. We added the following sentence to the discussion of Figure 2:*

Some sample plots for this day from the data of the three instruments can be found in the appendix.

**1.3 Detailed comments**

L15: Please add a reference to support your first sentence.
*The reference was at the end of the second sentence. We moved it to the first sentence.*

L16: Add comma after radiation
*done.*

L18: equatorward
*Corrected.*

L21: What do you mean by "stabilized"? some of these mechanisms could produce the dissipation of the clouds (e.g. CTEI).
*To clarify we end that sentence with a colon, and added a sentence explaining CTEI and refer to the discussion in Wood (2012).*

This system is stabilized by several feedback mechanisms: ... It has been hypothesized that evaporative cooling at cloud top may increase turbulent exchange between the moist boundary layer and the dry, free troposphere and thus dissipate the cloud (cloud top entrainment instability, CTEI). But it has been found that this mechanism requires special conditions and occurs less frequent than originally thought (Wood:2012).

L24: Parenthesis missing in the references.
*Corrected.*

L25: The subject of the sentence is "stratocumulus" which is a singular noun, hence the correct verb is "has"
*Corrected.*

L28: ... and, accordingly, it determines...
*Corrected.*

L44: Replace "turbulence" by "eddies".
*Replaced.*

L46: ... entrainment, a too low ...
*Corrected.*

L64: Eliminate parenthesis around references. Replace ; by and
*We follow here the rules of the publisher Copernicus: "If the author's name is not part of the sentence, name and year are put in parentheses" and "If you refer to multiple references at the same position all references are put in parentheses separated by semicolons". Thus not changed.*

L73: ... waters, cloud cover...
*Applied.*

L79: Latitude and longitude are incorrect
*Sorry, they were indeed wrong. Corrected.*

L79: What is DFG?
*'Deutsche Forschungs Gemeinschaft' = German Research Foundation . Text adapted.*

L101: What do you mean by "larger parts of the coast"? Please rephrase.
*We rephrased to :*

This a somewhat unique setting at the coast of northern Chile, as the cliff usually drops from several hundred meters directly into the ocean with only narrow stretches of rocky beaches.

L114: Remove extra parenthesis
*Removed.*

L134: The authors are too vague in describing the expected performance of the instrument: "to some extent" and "coarse vertical resolution". Please be more specific and quantitative.
*We removed this vague wording, the sentence is now:*

From these brightness temperatures integrated water vapor (IWV), liquid water path (LWP), temperature profiles and water vapor profiles are derived.

*In the following paragraph about the temperature profile we added the sentence:*

Water vapor profiles are vertically coarsely resolved using MWR measurements, i.e. there are only approximately two independent pieces of information contained in the TBs for water vapor profiling.

L142: How site-specific is the calibration methodology of the MWR? The fact that you use radiosondes far from Iquique to perform the calibration introduces uncertainty in your results? Please comment.
*The actual absolute calibration of the MWR TBs is performed with liquid nitrogen as mentioned in the first paragraph of 2.2.2. We use a climatology of radiosondes from Antofagasta (about 400 km to the south of Iquique) to derive regression coefficients for multi-variate retrievals of liquid water path, integrated water vapor and temperature and humidity profiles. Since both radiosonde stations are basically located at sea-level, we do not expect any systematic offsets due to differences in pressure broadening of the absorption lines. In addition, the linear relationship between the TBs in the K-band and humidity is largely independent of temperature, respectively between the TBs in the V-band and temperature is largely independent of humidity, so that we do not expect any significant systematic differences when applying the Antofagasta retrieval coefficients to the TBs measured in Iquique.*

L143-154: I believe this paragraph is better put at the end of section 2.2.2
*We agree, we moved it to the end of this section.*

L163: Please be explicit and quantitative on the vertical resolution attained by the instrument.
*We adapted the text about vertical resolution:*

An intrinsic feature of passive remote sensing is that spatial resolution

decreases with distance. As a result the inversion appears much broader than it is in reality, and it becomes impossible to identify bottom and top of the inversion layer. It will be also difficult to identify a moist adiabatic temperature profile in a cloud layer of a few hundred meters directly below the inversion.

L190: It would be nice if you can show the position of these 2 points in Figure 1.

*Done.*

L205: I was confused by the terminology "Boundary layer classification". I was expecting different classes of BL, but even in 1 BL case the scheme classifies different layers of the BL in different categories. Hence I believe a better name would be "turbulent layers classification" or "turbulence classification".

*Of course one could discuss the naming of this classification scheme:*
*Is a convective boundary layer with mixing driven by surface heating a different boundary layer than a stratocumulus capped boundary layer where mixing is driven by processes at its top (both are mixed by large plumes). Or is their turbulence basically different: skewness has a different sign but plumes move only in the opposite direction)?*
*To avoid this discussion we follow here just the name Manninen et al (2018) gave to this classification scheme.*

L206: Cloudnet

*Corrected.*

L208: Only below the cloud? I get the impression you included the cloud layer as well.

*Yes, only below the cloud as the Doppler lidar cannot see into the cloud.*

L239: Figure 2 includes a panel with the cloudnet classification for the day. However you do not discuss it at all. In particular, it is curious that this specific case does not show the "averaged" behavior of the dissipation. The cloud base does not rise much during the morning and at some time the cloud layer suddenly disappears (see major comment). I'd suggest that for this example day, you add a new figure with measurements of your 3 instruments and the cloudnet classification and expand the analysis and discussion. In this manner, the reader can assess how well the averaged fields presented later represent individual cases.

Figure 2: In the cloudnet panel please mark the times of the satellite pictures. The legend text is not readable.

*TO BE DONE*

L240-L260: As the authors indicate, winds in Figure 3 show a marked diurnal cycle and subtle seasonal variation. Therefore, I believe that the representation in Figure 3 could be enhanced. For example, show just 1 diurnal cycle (e.g. SON) and then show the difference of the diurnal cycles between 2 periods (JJA-DJF). This will make clearer the seasonal changes. Alternatively, show the annual diurnal cycle and the monthly variation of the 19-21 UTC and the 06-08 UTC winds corresponding to the extremes of the diurnal cycle.

*As the height of the detectable range varies with season the central feature of such a presentation is the difference in height. A scaling with the current detectable height range could solve this. But this generates ambiguities as this is either cloud base or top of the aerosol layer and may change from measurement to measurement. Although the idea is interesting we decided not to follow it.*

Figure 3: Please explain somewhere the reason of the reduced number of cases of some of the periods. Indicate if height is ASL or AGL. Also, please indicate whether all data in the figures is the average of the same number of days or the borders have less data. In the latter case, the robustness of the averages in the border should be discussed.

*We added a paragraph and a table at the end of the paragraph of the Cloudnet description:*

The instruments where equipped with battery backups to ensure controlled shutdown and recover in case of a power shutdown. Nevertheless this did not work in every case for every instrument and lead to several missing days. An instrument failure of the cloud radar lead to no data from end of May to begin of July. And damage of the radar radome forced us to shut down the instrument on January 22, 2019. As the Cloudnet retrieval can only be applied when all instruments provide data this lead to 243 days of data or 72% of the full year. Table 2 shows how many days are per month avaiable. There are few or no days with data in March, June and February and only 57% in January.

L262: Please indicate how the potential temperature was computed.

*We added the following text at the begin of section 3.3:*

Table 1: Number of days when all instruments could provide data to the Cloudnet retrieval during every month and season. Months are indicated by their first letter, $N$ is the number of days with data, $R$ is the data coverage relative to the days in the respective month or season in percent.

| Month | M | A | M | J | J | A | S | O | N | D | J | F |
|---|---|---|---|---|---|---|---|---|---|---|---|---|
| $N$/days | 6 | 25 | 27 | 0 | 23 | 29 | 30 | 30 | 29 | 27 | 17 | 0 |
| $R$/% | 19 | 83 | 87 | 0 | 74 | 94 | 100 | 97 | 97 | 87 | 57 | 0 |

| Season | MAM | JJA | SON | DJF |
|---|---|---|---|---|
| $N$/days | 58 | 52 | 89 | 44 |
| $R$/% | 63 | 57 | 98 | 49 |

We use the Temperature profiles from the microwave radiometer to calculate the potential Temperature with respect to the surface as $\theta = T_{\mathrm{air}} + \Gamma \cdot z$ with $\Gamma = g/c_P$. This is not exact but deviations are below 900 m typically well below 0.1 K.

Figure 4: Please show the temperature fields instead of the potential temperature fields. The most distinct feature of the thermal structure in the region is the prominent subsidence inversion which is most clear from temperature fields.

*We show potential temperature to visualize when and where the boundary layer is stable or neutrally stratified. This information would get lost when displaying air temperature. We consider this the more important information and therefore stay with potential temperature. As already described in section 2.2.2. about the microwave radiometer, the resolution of the temperature profile at boundary layer height is coarse and location and extension of the inversion should be regarded with caution. The strength of the temperature profile retrieval lies in the lower two thirds of the boundary layer.*

Figure 4: The same suggestions for Figure 3 apply here. Please explore ways of showing the seasonal and diurnal variation of the fields more distinctly.

*Same argument as above: Our focus is here to show stratification as a function of season and daytime. Temperature differences with respect to one season would make an interpretation in terms of stratification difficult. Additionally the variation of boundary layer height with season would become*

*a dominating feature in differential fields. That would require additional explanation how to interpret in terms of stratification. An alternative could be to use a normalized height but would require an additional plot with seasonal course of boundary layer height. We considered all these options already internal when developing this analysis and decided that this form of presentation is the best.*

Figure 4: Please indicate number of days used in the averages in this case.
*The analysis is restricted to the same days as the Cloudnet retrieval. We added a respective sentence in the figure caption.:*

Figure 4. ... The analysis here is restricted to the same days as the cloudnet retrieval.

L271: The height of the maximum temperature gradient should be compared with the average of inversion base and inversion top. In this case the comparison with the results of Munoz et al. 2011 is not so good, especially considering that IQQ is to the north of ANF. Please discuss.
*Indeed we falsely compared directly the height of the 'radiometer inversion' with the inversion base height $z_B$ in Muñoz et al 2011. We adapted the interpretation as follows:*

Despite the coarse resolution of the MWR this agrees with the annual course of the median inversion base height for the period from 1979-2007 in Antofagasta shown by Muñoz et al. (2011). Nevertheless we have to consider that cloud base, and thus probably also inversion height in Iquique, is on average by about 180 m higher than in Antofagasta (Muñoz et al. 2016). Additionally the inversion height data from the Antofagasta radiosonde (Muñoz et al. 2016) show beside a weak decrease of $-16$ m per decade, a large year to year variability in the order of $\pm100$ m. This makes it difficult to put our observations in relation to these long term analyses.

L276: The near surface stability is probably strongly conditioned by the land surface where the measurements are made and therefore they do not necessarily describe the conditions above the sea and the argument about the surface fluxes is weak. Please discuss.
*Lobos Roco et al (2018) did not refer to the near surface stability. We clarified this as follows:*

Lobos Roco et al. (2018) have shown that during stable stratification between their site at 1100 m ASL and the airport at 53 m ASL, typically no

clouds are present.

Figure 5: Please express the FOC as percentages in the figure so that the discussion in L280-281 is better followed.

*We replaced the figure by one which shows the frequency of occurence of clouds detected by Cloudnet with cloud-base <2 km asl during different seasons. We slightly adapted the discussion:*

The Cloudnet target classification scheme is used to investigate the frequency of occurrence (FOC) of warm clouds with cloud-base lower than 2 km asl (Fig.5). Most clouds occur in winter and spring, when during night about 90% of the time clouds are present, while during summer and autumn this reduces to 25% and 45%, respectively. During daytime FOC reduces by about 30%-points compared to the night meaning nearly no clouds during summer afternoon and somewhat more clouds in autumn and spring. In winter FOC stays allways above 60%.

L287-288: Please check whether this feature of the morning dissipation is due to the averaging or indeed happens in most of the days (see comment of Figure 2).

*We checked that: It occurs indeed on many but not all days. Those days when it does not follow this pattern it seems to be connected to advection (inferred from satellite images) or drizzling clouds. This pattern also agrees with Munoz et al 2016. The very regular diurnal course of the LCL (Fig.16) indicates that this is the mechanism,*

L298: Actually, I think the comparison with Munoz et al. (2016) is not as good. The authors should make the comparison in more detail and compare also their results with the operational measurements performed at Iquique airport (see major comment).

*We carafully reanalyzed the Munoz et al. (2016) dataset and revised the text:*

Munoz et al.2016 found in a 33 year dataset from Iquqique, centered around 1997, a similar pattern with cloud bases higher during daytime than during night. Nevertheless they found lower night to day amplitudes (50 m-75 m) and also cloud bases which were especially in autumn and winter by about 200 m-300 m higher hen we observed now. This difference can only partly explained by the trend of -100 m/15yr they found. From this comparison it seems as if this trend has accelerated.

L305: "water loss by evaporation at cloud-top". The authors must be careful about what they mean with this phrase. Do they refer to liquid water or to total water? In my view, entrainment at the top of the cloud layer does not imply total water loss for the MBL, as the entrained air is intruded into the MBL. Liquid water can be lost due to evaporation increased by the dry air entrained. It is not clear what exactly the authors mean here. Please explain and see also major comment.

*We mean here water loss into the free troposphere. We adapted the sentence:*

... water loss by detrainment and thus evaporation into the free troposhere at cloud-top ...

L310: It appears that figure 8 is not discussed in the text.
*That is true. Given the length of the text we decided to drop it.*

L314-315: The phrase about the uncertainty of the LWP retrieval is not clear. Please provide an explanation of the uncertainty mentioned (20-30 ug/m2). And if this is the case, then figure 9 should in some way convey that huge uncertainty in the plotted values. As it is now, it is misleading because the uncertainty appears to be described by the 4 retrievals which are almost identical.

*We have specified the LWP error discussion. Please note that we are discussing a random uncertainty of individual LWP observations, thus the uncertainties for seasonal means will be significantly lower. We have now added to the text:*

The theoretical retrieval uncertainty of LWP is determined to 10 $g/m^2$. Calibration and absorption model uncertainties can lead to systematic LWP offsets which can mount up to a maximum error of 20-30 $g/m^2$. These effects have been corrected for by applying a clear-sky offset correction, which is applied to all-sky observations. Thus, we expect a random uncertainty for individual values of not significantly more than 10 $g/m^2$, which further reduces for seasonal averages by a factor of $1/\sqrt{N}$. I.e. LWP uncertainty is dominated by the uncertainty of the radiosonde data preprocessing.

L316: It is noteworthy that in JJA the maximum LWP is found very early in the night and values decrease thereafter. Please discuss that.
*This is indeed an interesting feature. We added the following text:*

It is noteworthy that in winter (JJA) the maximum LWP is found in the

first half of the night and values decrease thereafter. This might be explained by occurrence of drizzle especially in the nights of August when during up to 40% of the time drizzle occurs. Drizzle removes liquid water from the cloud and eventually evaporates and thus removes liquid water from the cloud.

L319: Eliminate "The height of".
*Done.*

Figure 9. I'd have expected that TH80 retrieval always produced larger LWP than TH95, but this is not the case in the afternoon, especially winter and spring. Can you comment on that?

*The LWP retrieval is based on a multi-variate regression of the 7 K-Band TBs. These channels not only contain information on the LWP, but also water vapor. In TH80 and TH95 clouds are detected under different water vapor conditions. This leads to different regression coefficients and thus to different TB/LWP dependencies.*

Figure 9: Please indicate variability and/or uncertainty of these figures.
L331: frequency
*Corrected.*

L338: I suggest that the authors make an effort to analyze in more detail the drizzle occurrence, considering that they have so much data available. Scatterplots of day-to-day indices of drizzle, cloud depth and LWP are missed. For example, the simplest parametrizations of drizzle in Sc clouds is to define a threshold in maximum LWC above which drizzle begins. As the authors have estimates of LWP and cloud base and top heights, such relationships can be tested. I strongly suggest that they explore such relationships.

*These are of course interesting points and they would would give enough material for an independent paper. We included the section about drizzle to show that also this data is available. We keep the suggestion of the reviewer in mind for another publication.*

L343: See my concern with the "BL classification" terminology.
*As already said above: we stay with the name given by the authors.*

L361: What do you mean by "counter-intuitive"? I suggest the adjective "contrasting" or be more specific about the intuition behind the comment.
*It seems as if our intuition is different from the one of the reviewer, but maybe that is defined by our home climate zone. We changed the sentence to*

*:*

This seasonal difference in convective turbulence reflects the different stratification during the two seasons.

Figure 12: Can you separate your data in Cloudy/Clear nights and see how these diurnal classification changes? I believe that such analysis can provide more physical insights.

*This is would be a good idea but reviewer 2 already complained that the paper is too long. We keep it in mind for an upcoming analysis.*

L366-367: Again the argument of water loss by evaporation. Liquid water perhaps, but entrainment of tropospheric air into the MBL does not imply water loss for the MBL but the opposite. Please see major comment.

*As already argues entrainment of dry troposheric air into the MBL also include detrainment of moist air into the free troposhere.*

L376: in the cloud layer mixing produces moist-neutral conditions which are different than the dryneutral conditions in the mixed CBL.

*This is indeed the case, but it does not change the conceptual idea we want to bring forward here. We adapted the sentence to:*

Comparable to a CBL, intense turbulent mixing in the SCBL leads to neutral and moist-neutral stratification below the cloud and in the cloud, respectively.

L385: I miss a consideration of subsidence, especially if the authors are trying to describe some type of equilibrium condition. I believe no equilibrium will be attained if subsidence is not included in the analysis, as turbulent entrainment always tends to deepen the MBL. Also, in this section drizzle is not mentioned although it must be important in an equilibrium analysis.

*We added a description of the role of drizzle and subsidence:*

As long as the SCBL is well mixed and in a stable state, cloud-top energy- and mass-fluxes must be compensated by fluxes at its bottom. If these fluxes are not balanced, the SCBL will change its temperature and/or water content. Radiative emission at cloud top must be balanced by sensible heatflux from the ocean surface. Loss of water vapor by detrainment into the free troposphere must be compensated by evaporation at the ocean surface. Drizzle removes liquid water from the cloud. But as long as it does not reach the surface it does not change the water content of the MBL. A further factor is large scale subsidence. In a CBL warm rising plumes cross the inversion and

thus lead to a growth of BL height and this growth is reduced by subsidence (see e.g. Oversloot and Vila-Guerau de Arellano, 2013). In a SCBL this role of resistance against subsidence is taken by the entrainment of air from above the BL (Wood, 2012), nevertheless eintrainment rates remain a factor of uncertainty.

L407: There is ample literature on the relationship between lower-tropospheric stability and Sccapped MBL, which should be cited.

L410: Again the idea of water loss at the top of the MBL. See major comment.
*We dropped the sentence.*

L412: and again.
*We dropped that part of the sentence. The paragraph reads now:*

When the MBL has the same temperature as the ocean, plumes from cloud-top can reach the ocean surface and moisture from the ocean is mixed into the boundary layer and the cloud layer. In winter we have accordingly two mechanisms which support a persistent thick stratocumulus cloud deck: coupling of the MBL to the ocean surface and decoupling of the MBL from the free troposhere by a strong inversion. These winter conditions are accompanied by stronger large-scale subsidence caused by the then closer located center of the south-east pacific high pressure system. This subsidence further sharpens the inversion and slightly lowers its height (Fig.4) compared to summer. As a result the winter season coincides with times of most frequent cloud presence (Sect. 3.4, Fig. 6).

L417: and again.
*Not changed, as we believe that detrainement exist.*

L422: All the analysis in this section endeavors to show a cause-effect relationship between SST, subsidence and clouds. However, as the authors are discussing long-term averages I see a conceptual difficulty in establishing a cause-effect relationship. In my opinion, the most can be said is that there is an association between the SST, subsidence and cloudiness in these averages.
*We changed the begin of the last paragraph of this section to:*

We believe that our one year data show that ...

L425: Please see a previous comment on the "dissolution from the bottom" effect in the averages in relationship to the fourth panel of figure 2.

*We see that in many cases cloud base increases after sun rise. We therefore see no reason to make here any change.*

L427: Eliminate extra "in the"
*Removed.*

L434: What do you mean by "on the order of the values" ? what values?
*We simplified the sentence to:*

Nevertheless, day to day variability of IWV at every hour is large and these diurnal variations might be just random.

L436: Again the concept of water loss to the free troposphere.
*We changed the sentence to:*

Additionally, it must be noted that IWV is the column integrated water vapor which means that entrainment of dry air from the free troposphere does not change the IWV.

L437: Do you mean "winter"?
*Yes, we do. Corrected.*

L440: Please correct the "dew" subindex.
*Done.*

L441: The 400 m difference is huge and deserves more analysis. The 3.2 K surface super adiabatic condition needed to explain the difference conflicts with the cloudy conditions and the overall assumption of the paper that their measurements represent the coastal conditions over the sea surface. The authors should make an effort to compare their results with measurements at the airport (cloud base and surface variables), as well as LCL and cloud bases reported previously by Munoz et al. (2016).
*We changed the whole paragraph as already written in the reply to the major comment about such a comparison.*

L456: stratocumulus
*Corrected.*

L456: An alternative hypothesis for the daytime coastal clearing can be induced subsidence by the strengthening of the westerly component of the coastal wind during the day. I believe the data analysis of the authors is not sufficient to discard this hypothesis which has no need of an upper branch in the circulation. Their measurements of this circulation are only marginal as shown in figure 3.

*This is an interesting hypothesis. But wind lidar data show in cloud free moments the upper branch of the circulation. And we do not see significant changes of the (MWR-)inversion height, or cloud top height. Nevertheless it is possible that at the western end of the cloud gap subsidence plays a role.*

L462: The term "Rutllant cell" was coined in a paper by Houston.
*We adapted to :*

... the land-sea-breeze with the 'Rutllant cell' as desribed in Rutllant et al.(2003) ...

L462: "cannot be inferred"
*Corrected.*

L463: Replace "as it is " by "as the latter is"
*Done.*

L466: A scatter plot between LCL and cloud base would be of interest to appreciate their relationship beyond mean conditions.
*We replaced the lower right subplot by a joint pdf of LCL and cloudnet cloud base height, and adapted the discussion as described in the reply to the major comment at the begin.*

Figure 15: The quality of this figure is substandard. Some panels show the variability and others do not or do so partially. Ranges of the vertical axes are all different, which makes comparisons difficult.
*The range of the figures was indeed too small to show the full range of the standard deviation. We adapted the range of the figures.*

Figure 16 caption: two dimensional frequency distributions
*Adapted.*

---

## Author Comment (AC2)

**1 Reply to Reviewer 2**

Review of the article titled Life Cycle of Stratocumulus Clouds over one Year at the Coast of the Atacama Desert by Schween and coauthors for publication in the Atmos. Chem. Phys.

The authors have used 1-year of data collected at the airport site in Northern Chile to document the seasonal, and diurnal cycle of clouds, water vapor, LWP and turbulence. Focus is on marine boundary layer clouds. The article is overall well-written and will be of interest to the general meteorological community. Data in that part of the world is very rare, so the work is novel. Please find below my comments that can further improve the paper.

*We are grateful to reviewer 2 for his comments and suggestions to improve the paper. We will adress the points with italic text like this and provide changes in the text in red .*

**1.1 Major Comments:**

The paper is very long at this moment with 16 figures and 2 tables. I suggest you combine some of the figures and maybe put some in the supplemental material to reduce the paper. Figure 5 is redundant due to figure 6, so maybe put figure 5 in the supplemental material. Same thing can be done for Figures 12 and 13. You can also combine the Figure 6 and figure 7, by putting the cloud boundaries on top of the cloud fraction. Currently the paper is too long, and it will be good if you can bring it down to 10 figures. Thanks.

*We believe that Figure 5 (cloud FOC per hour) by itself provides valuable information which cannot directly be found in Figure 6 (cloud FOC per hour and height interval). An incorperation of Figure 7 (mean cloud base and top heights per hour) into Figure 6 (cloud FOC per hour and height) including the scatter would make Figure 6 very difficult to read. Figure 12 (BL class FOC per hour) and Figure 13 (BL class profiles) show different things and we see no way to combine their content. We therefore stay with the current selection of plots.*

Figure 14, 15 and 16 and the associated text, you have tried to probe largescale fields that might control the boundary layer dynamics and cloudiness. I suggest you plot the lower tropospheric stability (Klein and Hartmann, 1993) or Estimated Inversion Strength (Wood and Bretherton 2006). You can further plot all the reanalysis reported surface sensible heat flux and

latent heat flux. These quantities over the ocean and over the land site will tell you if there are any local factors that differ from the ocean and the site. This might also illuminate why the marine clouds evaporate over land at your site. Your explanation of stability and winds etc. ignores advection, and it can simply be the case that the clouds form over the ocean and dissipate over land due to lack of moisture supply from the surface, rather then shortwave heating.

*We think that an investigation of surface fluxes would be a good starting point for a further analysis. Nevertheless you have to consider that the topography at the coast is rather extreme with a steep cliff of 400 m height at just 4 km distance from the coastline. Further inland topography rises within a few kilometers to more than 1 km height asl, i.e. above the ocean inversion. The IFS has at the location of IQQ already an elevation of around 200 m above sea level, i.e. 150 m higher than in reality. Such an analysis would therefore require a careful validation of model or reanalyis data. We believe that this becomes too far out of the scope of this paper.*

Figure 15 is not in a suitable form. The standard deviation lines are not visible for any season except JJA.

*we apologize for that. Reviewer 1 had a similar comment. We adapted the scaling of the figure.*

I think it will be good if you plot the phase diagrams of surface winds to understand any local circulations. There are many papers on such a phenomenon, so not going to mention here. Please look at papers that probe the land-sea breezes. Probing this will make your article much stronger. Thanks.

*Focus of the paper are the clouds. The wind profile is more a supporting argument to explain the diurnal development of the clouds. We searched the literature but we found no representation or analysis which we thought would increase the understanding of the phenomenon. Given the fact that the paper is already too long and we were asked to reduce the number of figures we decided to leave it as is.*

Last major thing I will mention is the lack of information on profiles of turbulence. The Doppler Lidar was pointing vertically, so you can derive estimates of variance and skewness of vertical velocity. These are also used for PBL classification. I suggest you show the diurnal cycle of these quantities same as you have done for cloud properties.

*As described in section 2.3.2 the boundary layer classification scheme incorporates turbulent dissipation rate $\epsilon$ and vertical velocity skewness $S_w$. Turbulent dissipation rate is derived at every height from the spatial power spectrum of vertical velocity as described in Manninen et al. (2010). Figures 12 and 13 accordingly represent these properties with the distinction non turbulent versus turbulent based on $\epsilon \lessgtr 10^{-4} m^2 s^{-3}$ and the distinction cloud driven / convective based on $S \lessgtr 0$. We understand the interest of the reviewer to view profiles of turbulent properties and their development over the day. But again: the paper is already long and we were asked to reduce its length.*

**1.2  Minor Comments:**

It will be good if you show the diurnal cycle plots as a function of local time rather than UTC. This will make things easier to understand.

*We intentionally used UTC as time axis because for this location night appears at the left side and daytime on the right side. We incorporated in all plots the average time of sunrise and sunset to clarify this. We believe that this is the better method of representation as neither night nor day are split. Nevertheless we added a sentence to figure 3 to clarify:*

Night appears on the left side, and day on the right side of each plot.

Line 23-24: Mention precipitation loss of water too. Also, not sure what you mean by fresh. Thanks.

*"Fresh Water" at this point is indeed somewhat misleading. We included precipation as mechanism. the sentence reads now:*

Evaporation from the ocean and mixing through the boundary layer provides a continuous flow of water vapor balancing the water loss at cloud-top (Schubert et al. 1979I, Stevens et al. 2003) and precipitation back into the ocean (Wood 2012).

Line 39-41: These are very bold statements. So can you please add reference to support them? Thanks.

*The reference is the same as in the sentence before. We nevertheless added it at the end of the sentence.*

Figure 1: Not sure what is the point of showing cloud boundaries on this map. They are also difficult to identify and not discussed in the text.

*We added a sentence:*

The stratuscloud thus interferes with the topography whithin a few kilometers, provides as fog water to the surface and is limited in its extension to the east (blue and red lines in Fig.1).

Line 111: you mean eastern Pacific?
*Yes we do - we corrected that.*

Line 220: situation seems like a strange word to use here.
*That sounds indeed a bit weird - We dropped the word stratocumulus in front of the word 'situation'*

Line 231: do you mean evaporate the clouds? Dissolve has a solid into liquid connotation.
*Yes we do. We used the term consequently seven times, too many languages interfering with each other in my brain. Replaced them all by evaporate.*

Line 248-253: Seems that the text contradicts the figure. Can you please double check? The numbers dont seem to add up.
*We checked the numbers: they are correct.*

---

## Author Response (AR2)

**1 Reply to Editor**

*We received the remaining two reviewer comments forwarded to us by the editor and we tried to do our best to follow the suggestions.*

Editor:

Thank you for your revised manuscript. Reviewer #2 has expressed concern that you have not adequately addressed their major comments. In particular I am asking you to revisit their initial review and address two comments:

(1) 'I think it will be good if you plot the phase diagrams of surface winds to understand any local circulations. There are many papers on such a phenomenon, so not going to mention here. Please look at papers that probe the land-sea breezes. Probing this will make your article much stronger.'

*As already written in our former reply to this comment we tried some other representations than the velocity azimuth display with vectors and color shading as shown in figure 3, but we found nothing which brought a clearer representation and gave more insight. The editor agreed that our former reply to reviewer #2 and the discussion of the land-sea breeze covers this point sufficiently.*

(2) 'Last major thing I will mention is the lack of information on profiles of turbulence. The Doppler Lidar was pointing vertically, so you can derive estimates of variance and skewness of vertical velocity. These are also used for PBL classification. I suggest you show the diurnal cycle of these quantities same as you have done for cloud properties.'

*We added two further figures presenting means of vertical velocity standard deviation and skewness at the begin of section 3.7 "Turbulence". They have numbers 11 and 12, follwing figures are shifted backwards. We discuss these figures in the text as follows:*

Mean standard deviation of vertical velocity ($\sigma_w$, Fig.11) remains most of the time below 0.5 m/s. Only during daytime higher values develop from the ground and reach up to 300 m in summer and autumn and 750 m in winter and spring. During night time the upper half of the MBL shows especially in winter and spring slightly higher values.

The mean diurnal course of the skewness (Fig.12) shows distinct night and day patterns: During night skewness is mostly negative with lowest values in the upper half of the MBL, while during daytime skewness in the lower 2/3 of the MBL is clearly positive. Nighttime skewness is most negative in winter and spring. Daytime positive skewness reaches higher up in winter and spring than in summer and autumn. Most positive skewness occurs at heights in the range 200-300 m.

Vertical velocity in rising, as well as subsiding plumes is typically concentrated in narrow chimneys with high speeds surrounded by larger regions with only small velocities compensating for the mass transport in the plume cores. As a result regions with rising plumes show positive skewness and regions with subsiding plumes show negative skewness. Accordingly night time turbulence is driven mainly by subsiding plumes, and day time turbulence is driven by plumes rising from the surface. Nighttime as well as day time turbulence in winter and spring is more intense and reaches further down and further up, respectively, when compared to the other half of the year.

To further investigate how frequent which type of turbulences occurs and how far its influence reaches we use the boundary layer classification scheme of Manninen et al. (2018). ...